# Global cellular response to chemical perturbation of PLK4 activity and abnormal centrosome number

Johnny M Tkach[1], Reuben Philip[1,2], Amit Sharma[1], Jonathan Strecker[1,2†], Daniel Durocher[1,2], Laurence Pelletier[1,2]*

[1]Lunenfeld-Tanenbaum Research Institute, Sinai Health System, Toronto, Canada; [2]Department of Molecular Genetics, University of Toronto, Toronto, Canada

**Abstract** Centrosomes act as the main microtubule organizing center (MTOC) in metazoans. Centrosome number is tightly regulated by limiting centriole duplication to a single round per cell cycle. This control is achieved by multiple mechanisms, including the regulation of the protein kinase PLK4, the most upstream facilitator of centriole duplication. Altered centrosome numbers in mouse and human cells cause p53-dependent growth arrest through poorly defined mechanisms. Recent work has shown that the E3 ligase TRIM37 is required for cell cycle arrest in acentrosomal cells. To gain additional insights into this process, we undertook a series of genome-wide CRISPR/Cas9 screens to identify factors important for growth arrest triggered by treatment with centrinone B, a selective PLK4 inhibitor. We found that TRIM37 is a key mediator of growth arrest after partial or full PLK4 inhibition. Interestingly, PLK4 cellular mobility decreased in a dose-dependent manner after centrinone B treatment. In contrast to recent work, we found that growth arrest after PLK4 inhibition correlated better with PLK4 activity than with mitotic length or centrosome number. These data provide insights into the global response to changes in centrosome number and PLK4 activity and extend the role for TRIM37 in regulating the abundance, localization, and function of centrosome proteins.

**\*For correspondence:**
pelletier@lunenfeld.ca

**Present address:** †Broad Institute of MIT and Harvard, Cambridge, United States

**Competing interest:** The authors declare that no competing interests exist.

## Editor's evaluation

This study analyses the molecular pathways that lead to growth arrest in human cells with altered centriole number, caused by inhibition of PLK4, the master regulator of centriole duplication. The authors identify the ubiquitin E3 ligase TRIM37 as a key mediator of this growth arrest, but, in contrast to previous work, they find that growth arrest correlates better with PLK4 activity than with the duration of mitosis or centrosome number. The work extends the role for TRIM37 in regulating centrosome protein levels, distribution, and function, and will be of interest to researchers interested in centrosome and growth regulation.

## Introduction

The centrosome is a multi-protein complex that is the major microtubule organizing center (MTOC) of metazoan cells influencing microtubule-based processes such as cell division, ciliogenesis, signalling, and cell motility (*Conduit et al., 2015*). Each centrosome consists of two microtubule-based centrioles surrounded by pericentriolar material (PCM) (*Conduit et al., 2015*). Each cell inherits a single centrosome from the previous cell cycle and subsequent centriole duplication is restricted to a single round of replication (*Gönczy and Hatzopoulos, 2019*). Centriole duplication initiates at the G1/S transition and is largely a stepwise pathway dependent on the upstream kinase PLK4 (*Gönczy and*

*Hatzopoulos, 2019*). PLK4 is a low-abundance protein (*Fode et al., 1996*) that is recruited around the mother centriole (*Sonnen et al., 2013*; *Kim et al., 2013*). PLK4 kinase activity is regulated by interaction with STIL and CEP85 resulting in the recruitment of SASS6 to form the cartwheel of the nascent daughter centriole followed by the recruitment of centriole elongation factors (*Moyer et al., 2015*; *Ohta et al., 2014*; *Liu et al., 2018*; *Schmidt et al., 2009*; *Kohlmaier et al., 2009*; *Tang et al., 2009*; *Comartin et al., 2013*; *Azimzadeh et al., 2009*). PLK4 functions as a homodimer and autophosphorylates itself in trans to generate a phosphodegron sequence that limits its abundance. This sequence is recognized by SCF$^{\beta\text{-TrCP}}$ and targets PLK4 for ubiquitin-mediated proteolysis (*Rogers et al., 2009*; *Cunha-Ferreira et al., 2009*; *Guderian et al., 2010*).

PLK4 misregulation is often associated with pathological states. Centrosome amplification is a hallmark of tumor cells (*Chan, 2011*) and may play a role in generating chromosome instability (*Ganem et al., 2009*; *Silkworth et al., 2009*) and promoting cell invasiveness (*Arnandis et al., 2018*). In cell culture and mouse models, overexpression, inhibition, or deletion of PLK4 results in p53-dependent arrest (*Holland et al., 2012*; *Lambrus et al., 2016*; *Wong et al., 2015*; *Fong et al., 2016*; *Marthiens et al., 2013*; *Coelho et al., 2015*; *Vitre et al., 2015*; *Serçin et al., 2016*; *Hudson et al., 2001*). A series of CRISPR/Cas9 screens identified the p53 pathway members, p53, p21(CDKN1A), 53BP1 and USP28 (*Lambrus et al., 2016*; *Fong et al., 2016*; *Meitinger et al., 2016*), and the E3 ligase TRIM37 as component of this response (*Fong et al., 2016*; *Meitinger et al., 2016*). A recent screen for mediators of supernumerary centrosome-induced arrest identified PIDDosome/p53 and placed the distal appendage protein ANKRD26 within this pathway (*Evans et al., 2021*; *Burigotto et al., 2021*).

TRIM37 is an E3 ligase that has been associated with a myriad of cellular functions including gene expression (*Bhatnagar et al., 2014*), peroxisome maturation (*Wang et al., 2017*), various signalling pathways (*Zhu et al., 2020*; *Fu et al., 2021*; *Chen et al., 2020*; *Wang et al., 2018*), and centriole biology (*Balestra et al., 2013*; *Balestra et al., 2021*; *Meitinger et al., 2021*; *Yeow et al., 2020*). There is no consensus on how TRIM37 mediates these functions since its activities have been linked to mono-ubiquitination (*Bhatnagar et al., 2014*; *Wang et al., 2017*), poly-ubiquitination (*Zhu et al., 2020*; *Chen et al., 2020*; *Yeow et al., 2020*; *Meitinger et al., 2020*), and E3-independent functions (*Wang et al., 2018*) that result in changes in protein activity (*Bhatnagar et al., 2014*; *Wang et al., 2018*), localization (*Meitinger et al., 2016*; *Balestra et al., 2021*; *Meitinger et al., 2021*), and abundance (*Wang et al., 2017*; *Zhu et al., 2020*; *Chen et al., 2020*; *Yeow et al., 2020*; *Meitinger et al., 2020*). A number of centrosome-related TRIM37 functions have been described. In the absence of TRIM37, a collection of centriole proteins such as CNTROB, PLK4, CETN1/2, CP110 accumulate to form aberrant assemblies referred to as condensates or 'Cenpas' (centriolar protein assemblies) (*Balestra et al., 2021*; *Meitinger et al., 2021*). These structures are dependent on the presence of CNTROB, but it is unclear why TRIM37 might suppress their formation in normal cells. TRIM37 is part of the 17q23 amplicon present in approximately 18% of breast cancer tumors (*Kallioniemi et al., 1994*) and overexpression of TRIM37 in these lines renders them sensitive to the PLK4 inhibitor centrinone (*Yeow et al., 2020*; *Meitinger et al., 2020*). TRIM37 interacts with PLK4 and CEP192 (*Meitinger et al., 2020*). Although TRIM37 can promote ubiquitination of PLK4, it is distinct from SCF$^{\beta-TRCP}$ modification since it does not result in changes to PLK4 abundance (*Meitinger et al., 2020*). In contrast, transiently overexpressed TRIM37 leads to CEP192 ubiquitination and its subsequent degradation (*Yeow et al., 2020*; *Meitinger et al., 2020*). In this model of TRIM37 function, PLK4-nucleated condensates consisting of PCM components facilitate mitosis in the absence of TRIM37 and the overexpression of TRIM37 decreases the cellular levels of CEP192 rendering cells sensitive to the loss of centrioles.

Here, we sought to determine how growth arrest is initiated in response to alterations in PLK4 activity and centrosome number. Using the specific PLK4 inhibitor, centrinone B, we modulated PLK4 activity to generate supernumerary centrosomes or centrosome loss and performed genome-wide chemical genetic screens in RPE-1 and A375 cells. Our screens identified distinct pathways mediating the response to partial and full PLK4 inhibition. Intriguingly, TRIM37 was required for growth when PLK4 was partially or fully inhibited but was dispensable for arrest triggered by PLK4 overexpression. Moreover, TRIM37 growth arrest activity was partially independent of its E3 ligase activity. These results highlight the complex role of TRIM37 and regulators of its function in the control of centrosome number homeostasis.

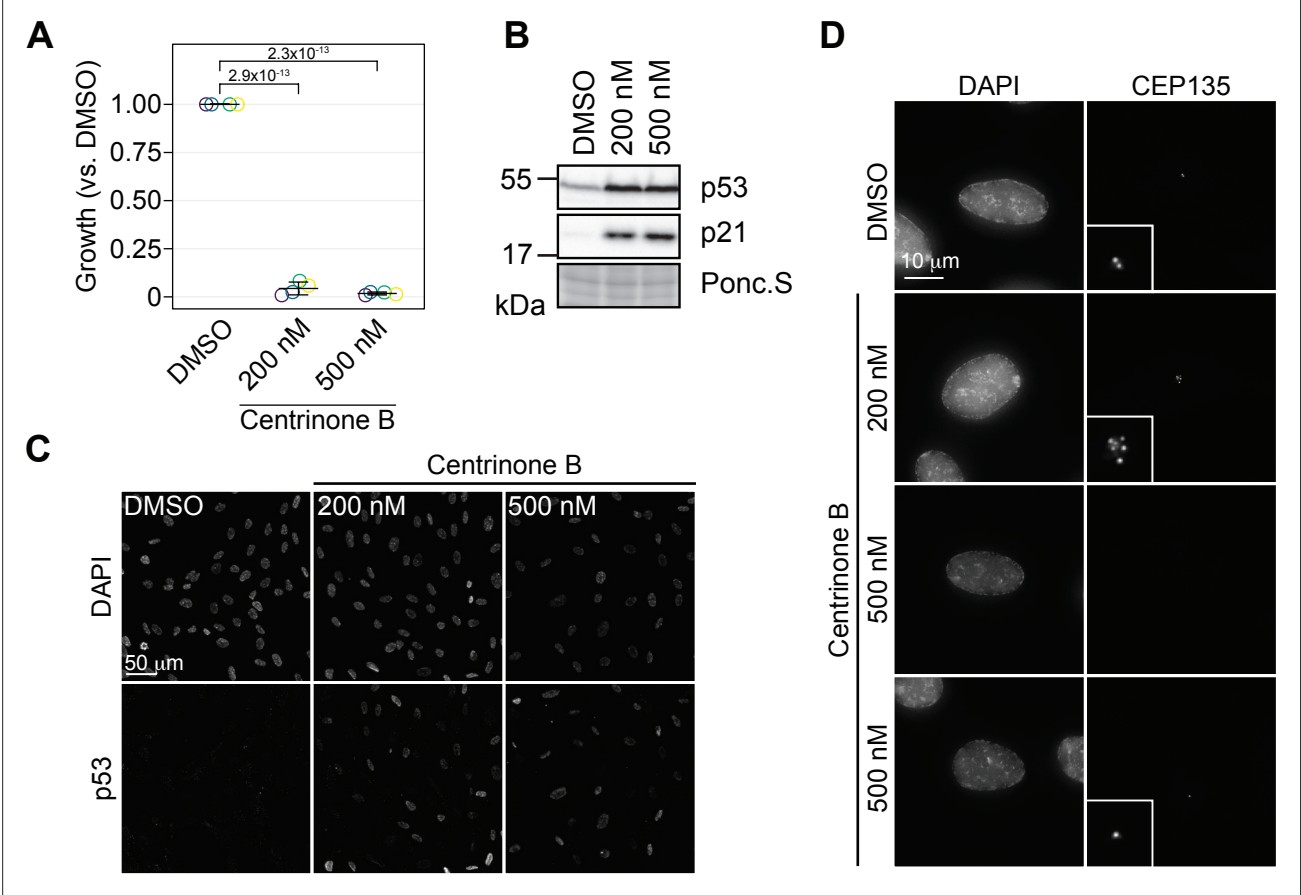

**Figure 1.** Concentration-dependent effect of centrinone B on centriole number. (**A**) RPE-1 cells were serially grown for 12 days and treated with DMSO or the indicated concentration of centrinone B. Relative cell number compared to a DMSO-treated control was determined and plotted. Three independent replicates plotted with mean with standard deviation shown. Significant *p*-values (<0.05) from Dunnett post hoc test using 'DMSO' as control after one-way ANOVA shown. (**B**) RPE-1 cells were treated with DMSO or 200 or 500 nM centrinone B for 4 days and prepared for Western blot probing for the indicated proteins. Ponc.S indicates total protein. (**C**) RPE-1 cells were treated as in (**B**), fixed for immunofluorescence and stained for p53. (**D**) RPE-1 cells were treated as in (**B**), fixed for immunofluorescence and stained for CEP135. Examples of cells with no centrosomes or one centrosome are shown. Inset magnified 3×. See *Figure 1—source data 1*.

The online version of this article includes the following source data and figure supplement(s) for figure 1:

**Source data 1.** Source data for *Figure 1*.

**Figure supplement 1.** Characterization of growth arrest and centriole abnormalities after PLK4 inhibition.

**Figure supplement 1—source data 1.** Source data for *Figure 1—figure supplement 1*.

## Results

### Centrinone B induces concentration-dependent changes in centrosome number

Centrinone is a PLK4-specific inhibitor that is often used to study the cellular response to centrosome loss (*Wong et al., 2015*). Another PLK4 inhibitor, CFI-400945, induces both centrosome amplification, or loss, in a concentration-dependent manner (*Mason et al., 2014*). Since CFI-400945 also inhibits other mitotic kinases such as AURKB (*Suri et al., 2019*) it is unclear if its reported phenotypes are due to effects on PLK4, mitosis, or both. Centrinone and centrinone B are both potent inhibitors of PLK4 that show selectivity over the Aurora-family kinases (*Wong et al., 2015*). We used centrinone B for our experiments since it shows an even greater selectivity over the Aurora kinases compared to centrinone (*Wong et al., 2015*; *Suri et al., 2019*). We treated RPE-1 cells with 200 and 500 nM centrinone B and found that cell growth was greatly inhibited at both centrinone B concentrations (*Figure 1A*). As expected, cell growth arrest was correlated with induction of p53 and p21 (*Figure 1B*,

*C*, *Figure 1—figure supplement 1A, B*), resulting in the accumulation of cells with 1 N DNA content (*Figure 1—figure supplement 1C*). We next imaged cells treated with centrinone B for 3 days to determine how centrosome number was affected as a function of centrinone B concentration. As expected, 500 nM centrinone B resulted in cells containing either a single or no centrosome, but cells treated with 200 nM failed to lose centrosomes and instead accumulated supernumerary centrosomes in approximately 50% of cells (*Figure 1C and D*, *Figure 1—figure supplement 1B, D*). Staining for a panel of centriole and centrosome markers revealed that these extra structures contained CEP135, CEP120, CETN2, glutamylated tubulin, and could accumulate PCNT and CEP192 indicating that treatment with 200 nM centrinone B can induce amplification of bona fide centrosomes (*Figure 1—figure supplement 1E*). Together, these data indicate that centrosomes can be amplified or lost using 200 or 500 nM centrinone B, respectively, and that both phenotypes result in p53-dependent G1 arrest.

## Global cellular responses to abnormal centrosome number

To understand the mechanisms of centrinone-induced cell cycle arrest, we performed genome-wide CRISPR screens in the presence of 200 and 500 nM centrinone B (*Figure 2A*; *Hart et al., 2015*). We reasoned that cell fitness at each centrinone B condition would require distinct sets of genes. Since loss of components in the p53 pathway itself would also increase fitness, we performed a parallel screen in the presence of Nutlin-3a, a small molecule that prevents the MDM2-mediated inactivation of p53 (*Vassilev et al., 2004*), allowing us to filter out core p53 pathway components. RPE-1 or A375 cells stably expressing Cas9 were infected with the TKOv1 lentiviral sgRNA library (*Hart et al., 2015*), selected, and subsequently treated with the indicated drug concentrations (*Figure 2A*). After growth for 21 days, cells were harvested and subjected to next-generation sequencing (NGS) and model-based analysis of genome-wide CRISPR-Cas9 knockout (MAGeCK) analysis (*Li et al., 2014*). Genes with a false discovery rate (FDR) of <0.05 were considered hits for subsequent analyses. The combined screens to identify regulators of growth arrest using Nutlin-3a, 200 and 500 nM centrinone B yielded 91, 136, and 56 high-confidence hits that positively or negatively affected cell growth, respectively (*Supplementary file 1*).

We created a network diagram to visualize the hits on a global scale (*Figure 2B*). Each unique cell line and condition act as the hubs (i.e., RPE-1, 200 nM centrinone B) while the hits from each condition are the remaining nodes. Each screen identified overlapping and distinct sets of genes, supporting our hypothesis that cells respond differently to each of the conditions tested (*Figure 2B*, *Figure 2—figure supplement 2A-D*, *Supplementary file 2*). The Nutlin-3a dataset contained core p53 pathway genes *TP53* and *CDKN1A* and genes coding for p53 regulators *TP53BP1* and *USP2*8, consistent with their role in promoting p53 transcriptional activity (*Cuella-Martin et al., 2016*). This dataset likely contains other mediators of the p53 pathway. The disruption of both *BAG6* and *EP300* increased fitness in Nutlin-3a. EP300 is an acetyltransferase that binds to and affects the acetylation of p53 while BAG6 modulates this acetylation event by EP300 (*Sebti et al., 2014*; *Liu et al., 1999*). Likewise, inactivation of *AHR* and *ARNT*, which form a transcriptional complex activated by exogenous ligands, promoted growth when p53 is activated in RPE-1 cells, but not A375. ARNT was previously identified as a FRET interactor with p53 (*Li et al., 2017*) and interacts with EP300 (*Tohkin et al., 2000*; *Tong et al., 2016*). Deletion of *TSC1/2* that integrates p53 signalling with the mTOR pathway (*Armstrong et al., 2017*; *Lee et al., 2007*) caused decreased fitness after p53 activation. We used Genemania (*Franz et al., 2018*) to further probe the pairwise physical interactions among the hits from the RPE-1 Nutlin-3a screen and generated a significantly enriched network (~20-fold enrichment, p=4.1 × 10$^{-31}$). In this network, 27 of the 57 hits formed physical interactions with eight proteins forming complexes with p53 itself (*Figure 2—figure supplement 1E*). Our high-confidence hits from the Nutlin-3a screen identified known p53 pathway members and likely contains unknown regulators of this pathway that will warrant further characterization.

The 200 nM centrinone B screens (i.e., condition that produces supernumerary centrosomes) revealed a core set of 23 genes that suppressed the growth arrest in both cell lines (*Supplementary file 1*, *Figure 2—figure supplement 2C*). Notably, we identified the ANKRD26/CASP2/PIDD1/CRADD (PIDDosome) complex recently implicated in the response to supernumerary centrosomes (*Evans et al., 2021*; *Burigotto et al., 2021*). This set also included p53 pathway genes (*TP53*, *CDKN1A*, *TP53BP1,* and *USP28*), centriole duplication factors (*CEP85*, *CENPJ*, *STIL*, *USP9X*), centriolar satellite proteins (*C2CD3*, *CEP350*, *KIAA0753*, *PIBF1*), and *TRIM37*. The A375 screen identified additional

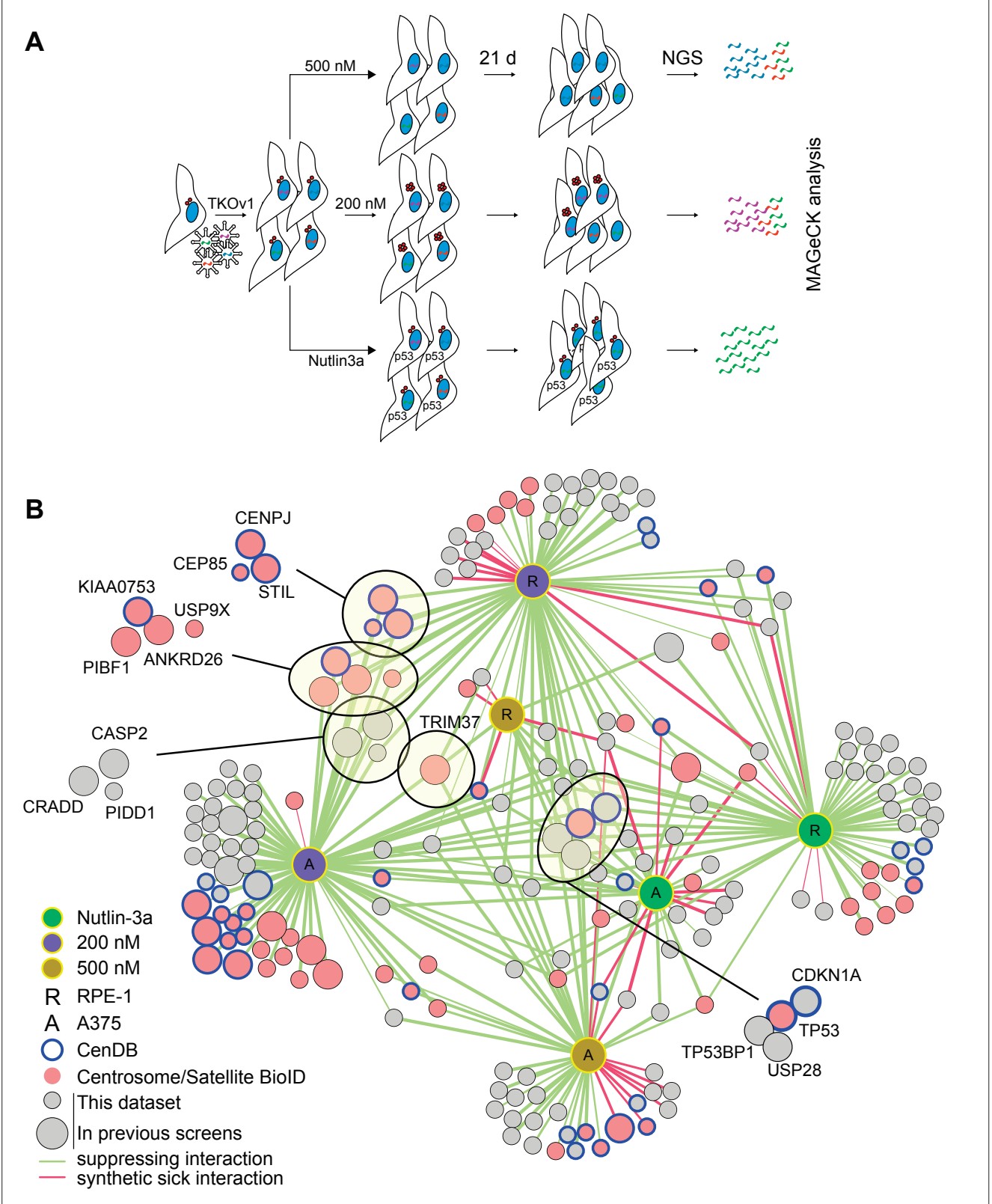

**Figure 2.** CRISPR/Cas9 screen to interrogate response to abnormal centrosome number. (**A**) Schematic outlining our screening procedures. Cells expressing Cas9 were infected with the TKOv1 genome-wide CRISPR sgRNA library and subsequently grown for 21 days in the presence of DMSO, 200 nM centrinone B, 500 nM centrinone B, or 600 nM Nutlin-3a. Genomic DNA was prepared and sgRNA counts in each pool of cells were determined using next-generation sequencing (NGS) and analyzed using model-based analysis of genome-wide CRISPR-Cas9 knockout (MAGeCK). Screens

*Figure 2 continued on next page*

*Figure 2 continued*

were performed in technical triplicate. (**B**) The significant hits (p<0.05) from all screens were combined to form a network. Each unique cell and drug combination used for screening were set as hubs (i.e., RPE-1 200 nM centrinone B). All other nodes represent genes identified. Edges connect identified genes with a screening condition with edge weight inversely proportional to false discovery rate (FDR). The general layout using the automated yFiles organic method from Cytoscape was preserved while individual nodes were manually arranged to facilitate visualization. Selected complexes and protein nodes are circled and highlighted. Except for the hubs, large nodes represent genes identified by previous PLK4 inhibition screens (see *Supplementary file 1*).

The online version of this article includes the following figure supplement(s) for figure 2:

**Figure supplement 1.** Bioinformatic analysis of CRISPR/Cas9 screens.

**Figure supplement 2.** Comparison of PLK4 screens.

centriole components namely *CP110*, *CEP97*, *CEP135*, *SASS6*, *CEP76*, and *PLK4*. Twenty-two of the hits from both cell lines combined also overlapped with the high-confidence hits from a similar screen that induced centrosome amplification by overexpressing PLK4 (*Evans et al., 2021*). The overexpression screen also yielded the ANKRD26/PIDDOsome and some, but not all the centriole-associated genes, nor *TRIM37* (*Figure 2—figure supplement 2F*).

We identified a total of 37 suppressors probing the response to centriole depletion (500 nM centrinone B), with five scoring in both cell lines (*TP53*, *CDKN1A*, *TP53BP1*, *USP28*, and *TRIM37*). Our results are similar to previous screens aimed at identifying suppressors of growth arrest due to centrosome loss from PLK4 inhibition that identified 31, 41, and 27 genes, respectively (*Lambrus et al., 2016*; *Fong et al., 2016*; *Meitinger et al., 2016*). Four of the five common hits (*TP53*, *TP53BP1*, *USP28*, *CDKN1A*) correspond to core p53 pathway components and were the only genes identified by all screens performed to date (*Figure 2—figure supplement 2G*). *CHD8* and *FBXO42* were previously identified in the response to centrosome loss (*Fong et al., 2016*) and also scored in our 200 and 500 nM centrinone B screens, respectively; however they also appeared in our Nutlin-3a hits suggesting that these genes might not be specific to centrosome biology. Indeed, both FBXO42 and CHD8 are known to negatively regulate p53 activity (*Sun et al., 2009*; *Nishiyama et al., 2009*; *Lü, 2022*). Two of the previous centriole loss screens also identified *TRIM37* (*Lambrus et al., 2016*; *Meitinger et al., 2016*) which was unique among all the other hits since it was the only gene outside the p53 pathway that scored in both RPE-1 and A375 cells in both centrinone B concentrations. We therefore chose to study TRIM37 further.

## TRIM37 localizes near the centrosomes but is not required for centriole duplication

Our combined screens indicated that TRIM37 was required for growth arrest in response to PLK4 inhibition that results in either centrosome overduplication or loss. Since TRIM37 has been implicated in centriole duplication (*Balestra et al., 2013*), we determined if loss of TRIM37 affected centrosome number after treatment with centrinone B. RPE-1 cells were infected with viruses expressing two independent sgRNAs targeting *TRIM37* (*Figure 3—figure supplement 1*) and selected cell pools were treated with 200 and 500 nM centrinone B and assessed for centriole number (*Figure 3A*). As previously observed (*Balestra et al., 2013*), we noted a small number of *TRIM37*-disrupted cells harbored extra centrioles in untreated cells. However, centriole amplification or loss after centrinone B treatment was not greatly affected (*Figure 3A*). To further characterize TRIM37, we created a *TRIM37*-disrupted RPE-1 clonal cell line using CRISPR/Cas9 (*Figure 3—figure supplement 1*). After treatment of cells with centrinone B, p53 and p21 failed to accumulate in *TRIM37*[-/-] cells at 200 or 500 nM centrinone B (*Figure 3B*). In the presence of supernumerary centrosomes, but not in the absence of centrosomes, MDM2 is cleaved via the ANKRD26/PIDDosome pathway that relieves its p53 inhibitory function and promotes p53 transcriptional activity (*Evans et al., 2021*; *Burigotto et al., 2021*; *Fava et al., 2017*). We also noted a small amount of cleaved MDM2 (*Fava et al., 2017*) in untreated *TRIM37*[-/-] cells with no additional increase after 200 nM centrinone B. This cleavage product was lost after treatment with 500 nM centrinone B. Thus, we find that TRIM37 does not affect gain or loss of centrosomes after centrinone B treatment but is required for the induction of both p53 and p21 in response to these treatments as previously observed (*Meitinger et al., 2016*), and is required for MDM2 cleavage.

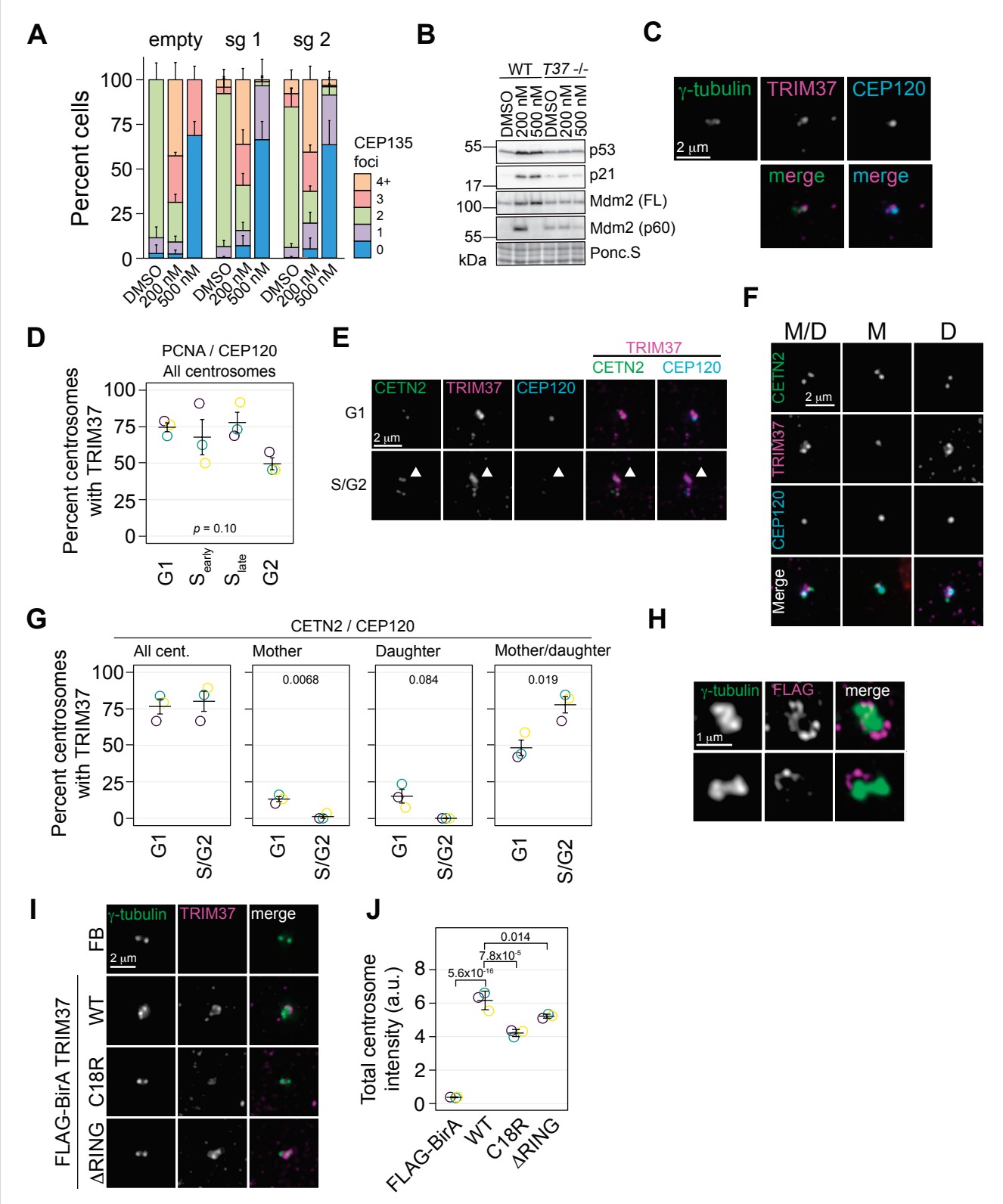

**Figure 3.** TRIM37 is a centrosome-associated protein. (**A**) RPE-1 Cas9 cells were stably infected with virus directing the expression of one of two sgRNAs against TRIM37 or empty vector. Selected cells were treated with DMSO, or 200 or 500 nM centrinone B for 4 days, fixed and stained for CEP135 and foci counted. Means and standard deviation shown (n=3, N≥169). (**B**) Cells from (**A**) were also processed for Western blotting using the indicated antibodies. FL – full length. p60 – p60 fragment. Ponc.S indicates total protein. (**C**) Asynchronous RPE-1 cells were fixed and stained with

*Figure 3 continued on next page*

*Figure 3 continued*

the indicated antibodies. Pairwise merged images are shown (bottom). (**D**) Asynchronus RPE-1 Cas9 cells were fixed and stained for TRIM37, PCNA, and CEP120. The number of TRIM37-positive centrosomes was manually determined for each cell cycle stage. Means from each replicate are shown as open circles. Resulting mean and standard deviation shown (n=3, N≥96). *p*-value from one-way ANOVA. (**E**) and (**F**) RPE-1 Cas9 cells were fixed and stained for the indicated antibodies. Examples of different cell cycles stages and TRIM37 localizations are shown in (**E**) and (**F**), respectively. Arrowhead in (**E**) indicates TRIM37 preference for one of two centrosomes. M/D: mother/daughter, M: mother, D: daughter. (**G**) Quantification of cells shown in (**E**) and (**F**). Individual data points shown as open circles. REsulting mean and standard deviation show (n=3, N = ≥94). Significant *p*-values (< 0.05) from a pairwise t-test between G1 and S/G2 populations indicated. (**H**) RPE-1 *TRIM37⁻/⁻* cells stably expressing FB-TRIM37 were fixed, stained with the indicated antibody, and imaged using 3D-SIM. Two representative images are shown. (**I**) RPE-1 *TRIM37⁻/⁻* cells stably expressing the indicated construct (FB = FLAG BirA) were pre-extracted, fixed, and stained for the indicated protein. (**J**) Centrosomal TRIM37 signal from cells in (**I**) was quantified. Means from each replicate are shown as open circles. Resulting mean and standard deviation shown (n=3, N≥84). Significant *p*-values (< 0.05) from Dunnett post hoc test using 'WT' as control after one-way ANOVA shown. Note that the results from (**I**) and (**J**) and those in *Figure 7—figure supplement 1C, E* are from the same experiment therefore 'FLAG-BirA' and 'WT' are duplicated in these panels. See *Figure 3—source data 1*.

The online version of this article includes the following source data and figure supplement(s) for figure 3:

**Source data 1.** Source data for *Figure 3*.

**Figure supplement 1.** Characterization of TRIM37 gene disrupted lines.

**Figure supplement 1—source data 1.** Source data for *Figure 3—figure supplement 1*.

**Figure supplement 2.** Characterization of clonal *TRIM37⁻/⁻* rescue lines.

**Figure supplement 2—source data 1.** Source data for *Figure 3—figure supplement 2*.

Since TRIM37 was a prey for multiple centrosome baits in our previous BioID survey of centrosomal proteins (*Gupta et al., 2015*), we sought to determine if TRIM37 localized to centrosomes. Immunofluorescence staining of endogenous TRIM37 indicated that the protein was associated with the centrosome in most cells although it did not strictly co-localize with either γ-tubulin or CEP120 (*Figure 3C*). We verified that the anti-TRIM37 antibody used reliably detected endogenous TRIM37. The overall TRIM37 signal was greatly reduced in TRIM37⁻/⁻ cells (*Figure 3—figure supplement 1*) and TRIM37 detected at the centrosome in interphase cells was largely diminished (*Figure 3—figure supplement 1*). We noted that a minor TRIM37 signal at the centrosome remained in *TRIM37⁻/⁻* cells but could not be further reduced using siRNA directed against TRIM37 (*Figure 3—figure supplement 1*) suggesting the antibody displays weak cross reactivity with another centrosomal component. Since the intensity difference in mitotic cells was not as large that observed in interphase cells after disruption and/or knockdown, we restricted our analysis to interphase cells (*Figure 3—figure supplement 1*). To determine if TRIM37 localization was cell cycle-dependent, we co-stained TRIM37 with PCNA and CEP120 (*Figure 3D*) and detected TRIM37 in all cell cycle stages we could discern. We further determined if TRIM37 preferentially localized to mother or daughter centrioles by staining with CETN2 to detect all centrioles and CEP120 that preferentially localizes to daughter centrioles (*Figure 3F and G*). TRIM37 localized to both mother and daughter centrioles in most cells and localized exclusively to the mother or daughter in only a small percentage of cells. Interestingly, in cells with two centrosomes, TRIM37 appeared to favor one centrosome over the other (*Figure 3E*, arrowhead) and we observed a minor preference for exclusive association with mother or daughter centrioles in G1 cells (*Figure 3G*). Given that the fluorescence signal from detecting endogenous protein was too low for super-resolution imaging, we performed 3D-SIM on RPE-1 Cas9 *TRIM37⁻/⁻* cells stably expressing FLAG-BirA-TRIM37 (FB-TRIM37) (*Figure 3H*, *Figure 3—figure supplement 2*). FB-TRIM37 formed partial ring structures preferentially surrounding one of the γ-tubulin foci. Moreover, the FB-TRIM37 signal was discontinuous with a dot-like distribution around the ring.

As a member of the TRIM family of E3 ligases, TRIM37 contains an N-terminal RING domain followed by a B-box and a coiled-coil region CCR (*Hatakeyama, 2017*). The E3 ligase activity of TRIM37 has been implicated in its centrosomal-related functions in mitotic length (*Meitinger et al., 2020*), PCM stability (*Yeow et al., 2020*; *Meitinger et al., 2020*), and PLK4 localization (*Meitinger et al., 2020*). We created two TRIM37 E3 ligase mutants, one containing a C18R point mutation (*Bhatnagar et al., 2014*) and another deleting the RING domain entirely (ΔRING). After stable expression in RPE-1 *TRIM37⁻/⁻* cells, the steady-state abundance of both ligase mutants was greater than the wild type (WT) protein, consistent with TRIM37 auto-regulating its stability (*Meitinger et al., 2020*; *Figure 3—figure supplement 2*). Correspondingly, immunofluoresence for the tagged proteins confirmed their relative abundances and demonstrated that the proteins were expressed in all cells

(*Figure 3—figure supplement 2*). The proteins were found primarily in the cytoplasm, sometimes in punctate structures possibly representing their peroxisomal localization (*Wang et al., 2017*). Further, we quantified the centrosomal localization of these proteins and found that both E3 mutants localized to the centrosome, albeit at slightly lower levels than the WT protein (*Figure 3I and J*). Thus, TRIM37 localizes to an area near the centrosomes proper, surrounding the PCM in a manner that is partially dependent on its E3 ligase activity.

## An E3-independent TRIM37 activity mediates growth arrest after PLK4 inhibition

To determine if TRIM37 E3 ligase activity promotes growth arrest after centrinone B treatment, we performed clonogenic survival assays with the E3 mutant rescue lines in the presence of DMSO, 200 or 500 nM centrinone B (*Figure 4A*). Expression of FB-TRIM37 fully restored the growth arrest triggered by centrinone B as did the expression of either C18R or ΔRING (*Figure 4A*). These data suggest that TRIM37 promotes growth arrest in response to centrinone B in an E3-independent manner.

To corroborate these observations, we created *TRIM37* knockout pools in RPE-1 and A375 cells using an sgRNA distinct from that used to make the clonal line (*Figure 3—figure supplement 1*). These pools were infected with virus to express TRIM37 and the indicated TRIM37 mutants (*Figure 4—figure supplement 1* and B). In addition, we performed clonogenic assays using varying centrinone B concentrations to fully characterize the growth arrest activity promoted by TRIM37 (*Figure 4B*, *Figure 4—figure supplement 1*). Robust cell arrest of WT cells was observed after treatment with 125 nM centrinone B or greater. Correspondingly, in RPE-1 cells, we observed increases of p53 and p21 abundance with increasing centrinone B that was attenuated in *TRIM37*$^{-/-}$ cells (*Figure 4—figure supplement 1*). Interestingly, cells lacking TRIM37 arrested after treatment with very low doses of centrinone B (50–125 nM) but only partially after higher concentrations (≥150 nM). As observed with cell lines derived from the clonal TRIM37 disruption, both E3-defective mutants, C18R and ΔRING, promoted growth arrest activity. To examine PLK4 function more closely after inhibition by centrinone B, we monitored centrosome number and cellular PLK4 mobility. There was a dose-dependent increase in centrosome number up to 167 or 200 nM after which cells harbored one or no centrosomes (*Figure 4C* and *Figure 4—figure supplement 1*). Surprisingly, abnormal centrosome number did not correlate with robust growth arrest in WT cells; cell growth was almost completely inhibited at 125 nM centrinone B although we observed minor centrosome abnormalities at this concentration. Recently, PLK4 phosphorylation status near its phosphodegron sequence was linked to its cellular mobility where decreased phosphorylation at this site resulted in decreased mobility (*Yamamoto and Kitagawa, 2019*). We expressed a PLK4 reporter construct (GFP-PLK kin + L1) and monitored PLK4 mobility using FRAP after treatment with centrinone B (*Figure 4D*; *Yamamoto and Kitagawa, 2019*). PLK4 mobility decreased with increasing centrinone B concentrations that mirrored growth arrest activity. Based on RPE-1 cells, our data uncovered three phases in response to PLK4 inhibition. Phase I is TRIM37-independent and occurs at centrinone B concentrations where cells display minor centrosome number aberration (≤125 nM); a TRIM37-dependent phase II occurs at ≥150 nM centrinone B and can be further separated based on centrosome number; phase IIa cells harbor overduplicated centrosomes (150–200 nM) while phase IIb cells have lost one or both centrosomes (500 nM) (*Figure 4E*). Similar trends were observed using A375 cells, although the exact concentrations of centrinone B required differed between the cell lines (*Figure 4—figure supplement 1*).

We noted that our FB-TRIM37 construct was overexpressed compared to the endogenous protein (*Figure 4—figure supplement 2A*), so we created *TRIM37*$^{-/-}$ cells lines that expressed inducible TRIM37-3xFLAG or TRIM37C18R-3xFLAG. Compared to our stable cell lines, the inducible lines expressed TRIM37 closer to endogenous levels, although the C18R mutant was always more highly expressed, similar to previous studies (*Figure 4—figure supplement 2E*; *Meitinger et al., 2020*). While the total cellular amount of the stable FLAG-BirA constructs was ~20-fold higher than that of the DOX-inducible system (*Figure 4—figure supplement 2*), the centrosomal difference was only ~3-fold (*Figure 4—figure supplement 2*). We performed clonogenic growth assays in the absence or presence of DOX and 0, 167, or 500 nM centrinone B (*Figure 4F*). Expression of either WT or C18R TRIM37 resulted in growth arrest although the growth arrest phenotype caused by TRIM37 C18R expression was weaker than that of WT cells. These data suggest that TRIM37 can support growth arrest after PLK4 inhibition in the absence of E3 ligase activity.

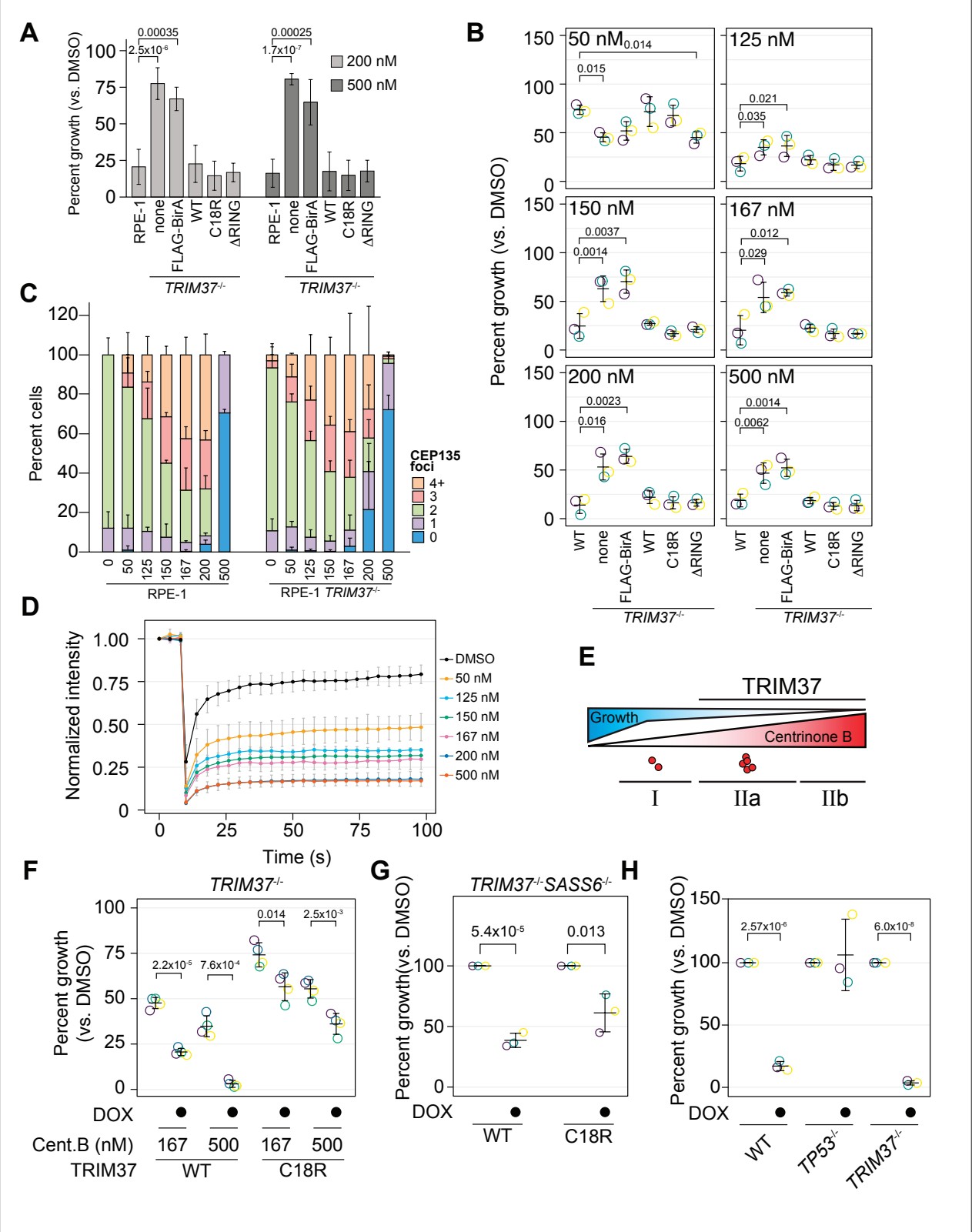

**Figure 4.** TRIM37 E3-independent activity is required for growth arrest. (**A**) WT RPE-1, *TRIM37*[-/-] (none), and *TRIM37*[-/-] cells expressing FLAG-BirA (FB) or the indicated FB-TRIM37 (WT, C18R, ΔRING) construct were seeded for clonogenic assays and grown in DMSO or the indicated concentration of centrinone B for 14 days. Colony density was quantified and growth compared to that in DMSO determined. Means and standard deviation shown (n=3). Significant *p*-values (< 0.05) from a Dunnett post hoc test using 'RPE-1' as a control after one-way ANOVA shown. (**B**) WT RPE-1, *TRIM37*[-/-]

*Figure 4 continued on next page*

*Figure 4 continued*

(pool), and *TRIM37⁻/⁻* expressing FB or the indicated FB-TRIM37 construct were seeded for clonogenic assays and grown in DMSO or the indicated concentration of centrinone B for 14 days. Colony density was quantified and growth compared to that in DMSO determined. Means from each replicate are shown as open circles. Resulting mean and standard deviation shown (n=3). Significant *p*-values (< 0.05) from Dunnett post hoc test using 'WT' as control after one-way ANOVA shown. Note that the results from this experiment and those in *Figure 7* are from the same experiment; therefore 'WT', '*TRIM37⁻/⁻* none', '*TRIM37⁻/⁻* FLAG-BirA', and '*TRIM37⁻/⁻* WT' are duplicated in these panels. (**C**) WT or *TRIM37⁻/⁻* (pools) RPE-1 cells were treated with DMSO (0) or the indicated concentration of centrinone B (nM) for 4 days before fixing and staining for CEP135. CEP135 foci per cell were manually counted. Mean and standard deviation shown (n=3, N≥55 per condition). (**D**) RPE-1 cells were transfected with GFP-PLK4kin + L1 and treated with DMSO or the indicated concentration of centrinone B for 16 hr. The mean and standard deviation among the independent replicates is shown (n=3, N≥12). (**E**) Model showing growth inhibition 'phases'. Growth is inhibited as a function of centrinone B. Phases dependent on TRIM37 are indicated. Red dots indicate centrosome number. (**F**) RPE-1 *TRIM37⁻/⁻* cells expressing DOX-inducible TRIM37-3xFLAG or TRIM37 C18R-3xFLAG were seeded for clonogenic assays in the absence and presence of doxycycline and DMSO or the indicated concentration of centrinone B. After incubation for 14 days, colony density was quantified and growth compared to that in DMSO determined. Means from each replicate are shown as open circles. Resulting mean and standard deviation shown (n=4). Significant *p*-values (< 0.05) from pairwise t-tests comparing -DOX and +DOX samples are shown. (**G**) RPE-1 *TRIM37⁻/⁻ SASS⁻/⁻* cells expressing DOX-inducible TRIM37-3xFLAG or TRIM37 C18R-3xFLAG were seeded for clonogenic assays in the absence and presence of doxycycline. After incubation for 14 days, colony density was quantified and growth compared to that in DMSO determined. Means from each replicate are shown as open circles. Resulting mean and standard deviation shown (n=3). Significant *p*-values (< 0.05) from pairwise t-tests comparing -DOX and +DOX samples are shown. (**H**) The indicated RPE-1 line expressing inducible PLK4-3xFLAG were seeded for clonogenic assays in the absence and presence of doxycycline. After incubation for 14 days, colony density was quantified and growth compared to that in DMSO determined. Means from each replicate are shown as open circles. Resulting mean and standard deviation shown (n=3). Significant *p*-values (<0.05) from pairwise t-tests comparing -DOX and +DOX samples are shown. See *Figure 4—source data 1*.

The online version of this article includes the following source data and figure supplement(s) for figure 4:

**Source data 1.** Source data for *Figure 4*.

**Figure supplement 1.** Cellular response to varying centrinone B treatments.

**Figure supplement 1—source data 1.** Source data for *Figure 4—figure supplement 1*.

**Figure supplement 2.** Comparison of protein abundance in stable and inducible cell lines.

**Figure supplement 2—source data 1.** Source data for *Figure 4—figure supplement 2*.

**Figure supplement 3.** Characterization of *SASS6⁻/⁻* and pInducer PLK4 cell lines.

**Figure supplement 3—source data 1.** Source data for *Figure 4—figure supplement 3*.

To distinguish between PLK4 activity and centrosome number, we used orthologous methods tTo corroborate these observations, we creato control centrosome number. First, we disrupted the gene encoding SASS6, a protein required for centriole duplication (*Leidel et al., 2005*; *Dammermann et al., 2004*), to induce centrosome loss in *TRIM37⁻/⁻* cells (*Figure 4—figure supplement 3A*). We recovered cells lacking centrioles indicating that TRIM37 is also required for growth arrest in these conditions since *SASS6* is essential in WT cells (*Leidel et al., 2005*). Interestingly, *TRIM37⁻/⁻SASS6⁻/⁻* cells were completely resistant to any dose of centrinone B, unlike *TRIM37⁻/⁻* alone (*Figure 4—figure supplement 3B*). Similar to centrosome loss induced by PLK4 inhibition, inducible expression of TRIM37 C18R partially rescued the growth arrest phenotype caused by SASS6 loss (*Figure 4G*, *Figure 4—figure supplement 3C,D*). To induce centrosome amplification, we overexpressed PLK4 (*Figure 4—figure supplement 3E*). Similar to previous studies (*Evans et al., 2021*), we found that disruption of *TRIM37* was unable to suppress growth arrest in these conditions (*Figure 4H*). Together, these data suggest that TRIM37 is required for growth arrest after moderate or full PLK4 inhibition and this activity is partially independent of TRIM37's E3 ligase activity.

## Growth arrest upon treatment with centrinone B does not correlate with an increase in mitotic length

Abnormal mitotic length has been put forth as an attractive model to explain growth arrest after centrosome loss (*Lambrus et al., 2016*). We measured mitotic length after inhibiting PLK4 at different centrinone B concentrations using live-cell imaging of WT or *TRIM37⁻/⁻* RPE-1 cells. Cells were treated with the indicated concentration of centrinone B for 3 days before imaging them for 24 hr, and the length of mitosis from nuclear envelope breakdown (NEBD) to telophase was quantified (*Figure 5A*). WT RPE-1 cells treated with centrinone B did not exhibit a significant increase in mitotic length, compared to untreated cells, until concentrations reached 150 nM. Strikingly, the mitotic length of *TRIM37⁻/⁻* cells treated with centrinone B was similar to that of WT cells until concentrations of 200

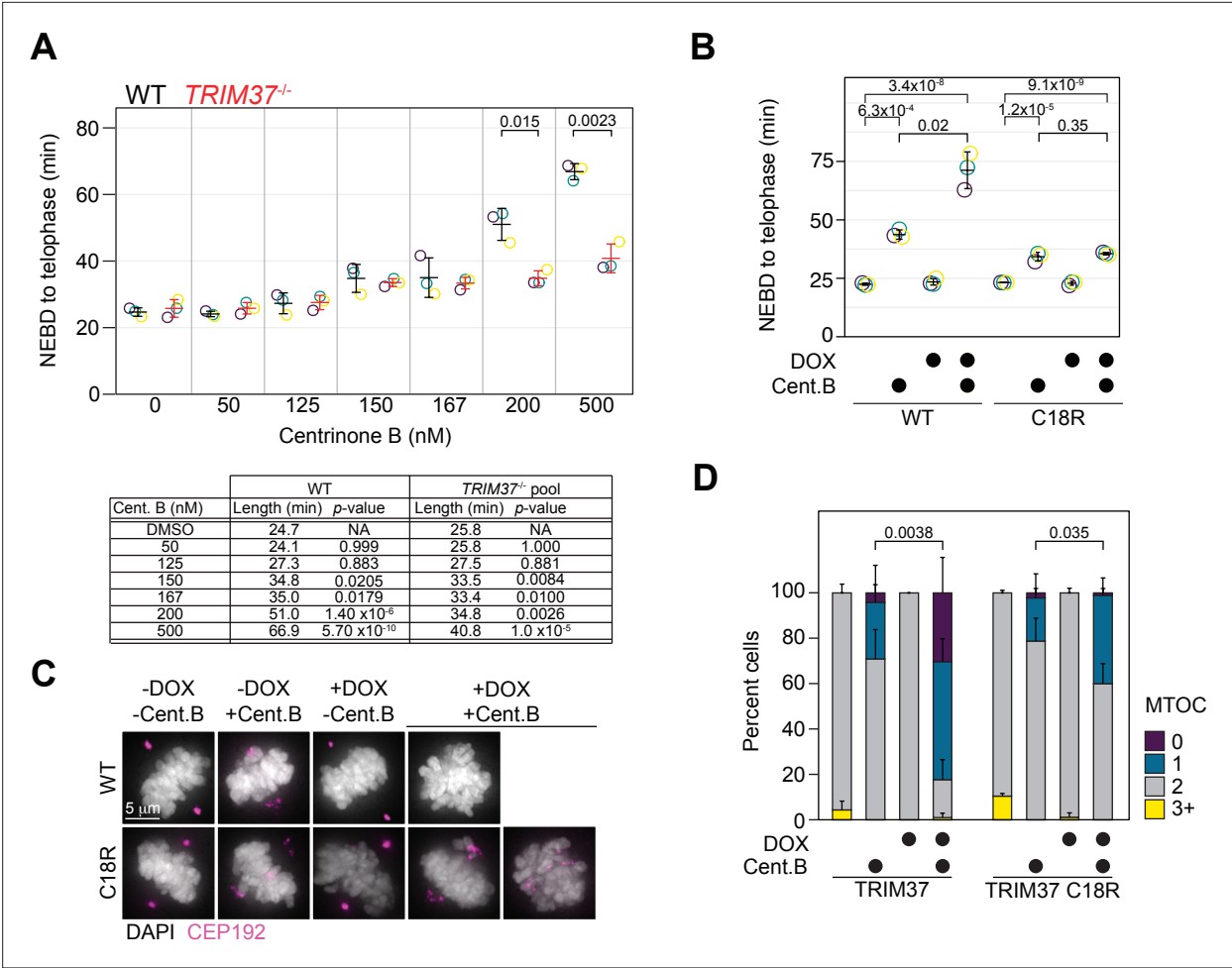

**Figure 5.** TRIM37 displays E3-dependent and -independent mitotic activities. (**A**) WT RPE-1 and *TRIM37*−/− cells were incubated with DMSO (0) or the indicated concentration of centrinone B for 3 days before live imaging for 24 hr. The time from nuclear envelope breakdown (NEBD) to telophase was determined. Means from each replicate are shown as open circles. Resulting mean and standard deviation shown (n=3). Significant *p*-values (< 0.05) from pairwise t-tests comparing WT and *TRIM37*−/− samples are shown. (n=3, N≥30). Table below indicates average mitotic length and p-value from a Dunnett post hoc test using 'DMSO' as control after one-way ANOVA. (**B**) RPE-1 *TRIM37*−/− cells were treated with DMSO or 500 nM centrinone B (Cent.B) in the absence or presence of doxycycline (DOX) for 3 days before live imaging for 24 hr. The time from NEBD to telophase was determined. Means from each replicate are shown as open circles. Resulting mean and standard deviation shown (n=3). Significant *p*-values (< 0.05) from Dunnett post hoc test using 'uninduced, DMSO treated' cells as a control after a one-way ANOVA are shown (n=3, N≥46). (**C**) RPE-1 *TRIM37*−/− cells expressing the indicated TRIM37 protein (WT or C18R) were treated with DMSO or 500 nM centrinone B (Cent.B) in the absence or presence of DOX for 3 days before fixing and staining for CEP192. Representative images shown. (**D**) The number of discernable microtubule organizing centers (MTOCs) characterized by the accumulation of CEP192 in cells from (**C**) was quantified. Means and standard deviation shown (n=3, N≥29). For each TRIM37 construct, the number of cells incubated with centrinone B and with two MTOCs in uninduced and induced samples was compared using a pairwise t-test. Significant *p*-values (< 0.05) are shown. See *Figure 5—source data 1*.

The online version of this article includes the following source data for figure 5:

**Source data 1.** Source data for *Figure 5*.

and 500 nM centrinone B; at which point the absence of TRIM37 partially suppressed the increase in mitotic length in WT cells (*Meitinger et al., 2020*). We next examined whether TRIM37 E3 activity was required to rescue the shortened mitotic length observed in *TRIM37*−/− cells. We induced the expression of WT TRIM37 and the C18R mutant in *TRIM37*−/− cells, treated them with 500 nM centrinone B and measured mitotic length. In the absence of protein induction, we observed a slight increase in mitotic length after centrinone B treatment and a larger increase after induction of WT TRIM37 (*Figure 5B*). In contrast, the mitotic length in the presence of centrinone B after expression of TRIM37 C18R did not change compared to similarly treated uninduced cells. Thus, the increase in mitotic

length observed after full PLK4 inhibition (500 nM centrinone B) appears to be dependent on TRIM37 E3 activity.

In the absence of centrosomes, amorphous collections of primarily PCM components act as pseudo-MTOCs in cells lacking TRIM37 (*Meitinger et al., 2016*). We induced the expression of WT TRIM37 or TRIM37 C18R in *TRIM37*[-/-] cells, treated them with 500 nM centrinone B for 3 days before staining for CEP192 (*Figure 5C*) and analyzed mitotic cells for the number of MTOCs in each cell based on CEP192 distribution (*Figure 5D*). In DMSO-treated cells, all lines mostly formed two distinct MTOCs. In the presence of centrinone B, but in the absence of induced protein, most cells formed two fragmented MTOCs, characteristic of *TRIM37*[-/-] cells after PLK4 inhibition (*Meitinger et al., 2016*). Cells expressing WT TRIM37 and treated with centrinone B harbored a single fragmented MTOC or none at all. Lastly, cells expressing TRIM37 C18R and treated with centrinone B displayed a partial phenotype where some cells formed two dispersed MTOCs while others displayed a single dispersed MTOC. Thus, TRIM37 E3 ligase activity is not strictly required to suppress pseudo-MTOCs or to promote cell growth after centrosome loss.

## TRIM37 affects the abundance and localization of both centriolar and PCM components in an E3-dependent manner

To assess whether TRIM37 regulates the abundance of centriolar or PCM components, we initially probed for PCM components in RPE-1 and RPE-1 *TRIM37*[-/-] lines but did not detect any significant changes in steady-state protein levels (*Figure 6A* and *Figure 6—figure supplement 1A*). To determine if overexpressed TRIM37 affected steady-state PCM levels, we again probed for PCM proteins in *TRIM37*[-/-] cells stably expressing FLAG-BirA, FB-TRIM37, or the TRIM37 E3 mutants. We observed a significant decrease in CEP192 that was E3-dependent but did not observe significant changes in PCNT or CEP215 (*Figure 6B* and *Figure 6—figure supplement 1B*). We next expressed inducible WT or C18R TRIM37 in WT RPE-1 cells (*Yeow et al., 2020*; *Meitinger et al., 2020*). After expression of WT TRIM37 for 4 or 8 hr, we detected a 50–75% decrease in total CEP192 protein (*Figure 6C*, left panels). In all cases the decrease in CEP192 was E3-dependent. These results are consistent with previous observations that also indicated that TRIM37 is a negative regulator of CEP192 abundance (*Yeow et al., 2020*; *Meitinger et al., 2020*). We extended these findings by using immunofluorescence to specifically detect changes at mitotic centrosomes (*Figure 6—figure supplement 1C*). Quantification revealed that acute expression of WT TRIM37, but not C18R decreased the intensity of CEP192 and PCNT at the centrosome (*Figure 6D*). Interestingly the centriole component CEP120 was similarly diminished at the centrosome indicating that this effect is not specific to PCM proteins (*Figure 6D*). We find that overexpression of TRIM37 affects the overall abundance of CEP192 and the mitotic accumulation of CEP192, PCNT, and CEP120 in an E3-dependent manner.

In the absence of TRIM37, some centriolar proteins form CTNROB-dependent ectopic intracellular aggregates, termed Cenpas, in interphase cells (*Balestra et al., 2021*; *Meitinger et al., 2020*). We initially observed that CEP120 was mislocalized in *TRIM37*[-/-] cells and co-localized with CNTROB (*Figure 6E*). Ectopic CNTROB structures remained after CEP120 depletion using siRNA suggesting that CEP120 is assembled downstream of CNTROB. (*Figure 6—figure supplement 2A*). We found that CEP120 and CNTROB were detected in these structures and CETN2 foci accumulated near them (*Figure 6E*). Notably, we did not detect PLK4 in the CNTROB/CEP120 structure. Previously, PLK4 has been observed in these structures but could only be detected using a single antibody and the signal remained after siRNA treatment (*Meitinger et al., 2016*; *Balestra et al., 2021*; *Meitinger et al., 2021*). To determine if PLK4 could be recruited into these structures, we expressed PLK4-3xFLAG from an inducible promoter in *TRIM37*[-/-] cells. PLK4-3xFLAG was not detected at non-centrosomal aggregates after 3 or 6 hr induction using either anti-FLAG or anti-PLK4 antibodies, despite its accumulation at the centrosome (*Figure 6—figure supplement 2B*). Additionally, these assemblies were not affected by the loss of PLK4 activity as they were observed in *TRIM37*[-/-] cells treated with 500 nM centrinone B for 3 days (*Figure 6—figure supplement 2C, D*). After stable expression of TRIM37 mutants in TRIM37[-/-] cells these structures disappeared in an E3-dependent manner (*Figure 6F*, *Figure 6—figure supplement 2E*). We find that CEP120 is a downstream component of the CNTROB structures formed in *TRIM37*[-/-] cells and that the suppression of their formation requires TRIM37 E3 ligase activity.

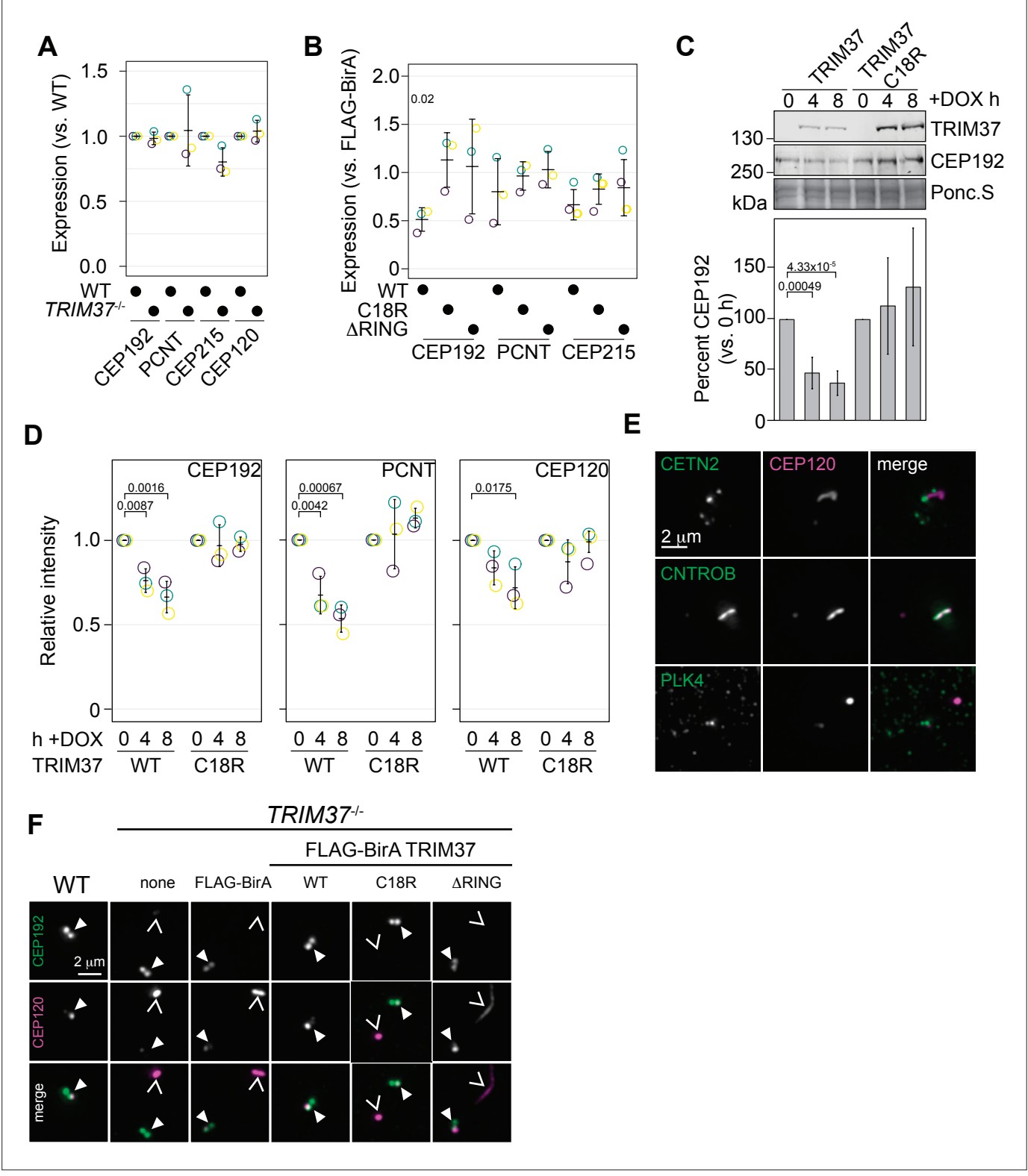

**Figure 6.** TRIM37 negatively regulates centriole and centrosome proteins. (**A**) WT RPE-1 and *TRIM37^-/-* cells were processed for Western blot and probed for the indicated proteins. Band intensity was quantified and expressed as expression compared to WT cells. Relative intensity from each replicate are shown as open circles. Resulting mean and standard deviation shown (n=3). Data was tested by pairwise t-test between WT and *TRIM37^-/-*. No significant differences were observed. (**B**) *TRIM37^-/-* RPE-1 cells stably expressing FLAG-BirA or the indicated FB-TRIM37 protein were processed for Western blot and probed for the indicated proteins. Band intensity was quantified and expressed as expression compared to cells expressing FLAG-

*Figure 6 continued on next page*

*Figure 6 continued*

BirA. Relative intensity from each replicate are shown as open circles. Resulting mean and standard deviation shown (n=3). Significant *p*-values (< 0.05) from Dunnett post hoc test using FLAG-BirA as a control after a one-way ANOVA are shown. The band intensity from FLAG-BirA cells was set to '1' and is omitted from the plot for clarity. (**C**) WT RPE-1 cells expressing doxycline-inducible TRIM37-3xFLAG (WT) or TRIM37 C18R-3xFLAG (C18R) were induced with doxycycline for 0, 4, or 8 hr. At each time point, extracts were prepared and analyzed by Western blot for the indicated protein (right). Ponc.S indicates equal loading. CEP192 abundance was quantified and normalized to the intensity at time 0 hr (bottom). Mean and standard deviation shown (n=3). Significant *p*-values (< 0.05) from Dunnett post hoc test using time 0 hr as a control after a one-way ANOVA are shown. (**D**) Cells from (**C**) were also fixed and immunostained for the indicated proteins. The centrosomal intensity from mitotic cells was determined. Intensity values were normalized to 0 hr. Means from each replicate are shown as open circles. Resulting mean and standard deviation shown (n=3, N=60). Significant *p*-values (< 0.05) from Dunnett post hoc test using time 0 hr as a control after a one-way ANOVA are shown. (**E**) RPE-1 *TRIM37*^-/- cells were fixed and stained for CEP120 and the indicated proteins. (**F**) RPE-1 (WT) or *TRIM37*^-/- cells stably expressing FLAG-BirA or the indicated FB-TRIM37 protein were fixed and stained for the indicated protein. Arrowhead indicates centrosome defined by CEP192. Caret mark indicates ectopic structure defined by CEP120. See *Figure 6—source data 1*.

The online version of this article includes the following source data and figure supplement(s) for figure 6:

**Source data 1.** Source data for *Figure 6*.

**Figure supplement 1.** TRIM37 negatively regulates pericentriolar material (PCM) and centriole proteins.

**Figure supplement 2.** Characterization of ectopic centrosomal aggregates.

**Figure supplement 2—source data 1.** Source data for *Figure 6—figure supplement 2*.

## TRIM37 promotes the phosphorylation of PLK4

TRIM37 is suggested to associate with and ubiquitinate PLK4 (*Meitinger et al., 2020*). We performed a structure-function analysis of TRIM37 to determine which region(s) of TRIM37 were required for PLK4 complex formation (*Figure 7A*) We transiently expressed a series of FB-TRIM37 deletion mutants and Myc-PLK4 in RPE-1 cells, immunoprecipitated the FB-TRIM37 constructs, and probed for PLK4 (*Figure 7A and B*, *Figure 7—figure supplement 1A*). Our results confirmed that PLK4 and TRIM37 form a complex in RPE-1 cells (*Meitinger et al., 2020*). Further, the region from amino acids 505–709 of TRIM37 was sufficient to immunoprecipitate PLK4. Conversely, a TRIM37 mutant lacking this region (FB-Δ505–709) failed to pull down PLK4. When stably expressed in *TRIM37*^-/- cells, FB-505–709 and FB-Δ505–709 were well expressed (*Figure 7—figure supplement 1B, D*) but only FB-Δ505–709 localized to centrosomes (*Figure 7—figure supplement 1C, E*). To determine if the PLK4/TRIM37 association was required for growth arrest activity, we performed clonogenic assays using *TRIM37*^-/- cell lines stably expressing FB-TRIM37 505–709 and FB-TRIM37 Δ505–709 (*Figure 7C*, *Figure 7—figure supplement 1B-E*) or expressing inducible FB-Δ505–709 (*Figure 7—figure supplement 1F, G*). Surprisingly, the association between PLK4 and TRIM37 did not appear to be required for centrinone B-induced growth arrest.

PLK4 protein abundance is tightly controlled by multiple post-translational modifications including phosphorylation and ubiquitination (*Rogers et al., 2009*; *Cunha-Ferreira et al., 2009*; *Guderian et al., 2010*; *Yamamoto and Kitagawa, 2019*). The co-expression of Myc-PLK4 and T7-TRIM37 in HEK293T cells resulted in modification of PLK4 (*Figure 7D*). The modification was partially dependent on TRIM37 E3 activity, was not observed when TRIM37 505–709 was expressed, and increased in the presence of TRIM37 Δ505–709. MLN4924 is a general inhibitor of cullin-RING E3 ligases and treating cells with this compound should inhibit ubiquitination of PLK4 by SCF$^{β-TrCP}$ (*Soucy et al., 2009*). Treatment of cells with MLN4924 resulted in stabilization of PLK4 but the modified forms remained (*Figure 7D*). To directly test if the observed modification was ubiquitinated PLK4, we co-expressed Myc-PLK4, T7-TRIM37, and HA-Ub in 293T cells. After immunoprecipitating PLK4, we probed for HA-Ub to detect ubiquitinated species (*Figure 7E*). Although we detected an E3-dependent increase in total ubiquitinated proteins in the input of cells expressing WT and Δ505–709 T7-TRIM37, we only detected low levels of HA-Ub conjugates in the anti-Myc immunoprecipitates, suggesting that PLK4 modification upon expression of TRIM37 may not be due to its ubiquitination. As an alternative possibility, we tested if the modified PLK4 bands were due to phosphorylation by treating cell lysates with $\lambda$-phosphatase (*Figure 7F*). The slower migrating forms of PLK4 were lost after phosphatase treatment indicating that these modifications are primarily due to phosphorylation. To identify the kinase(s) responsible for the modification, we treated cells with inhibitors targeting PLK4, AURKA, PLK1, and CDK1 for 3–6 hr and probed for PLK4. The only treatment that substantially reduced PLK4 phosphorylation was PLK4 inhibition (*Figure 7G*). Since we observed that TRIM37 promotes PLK4

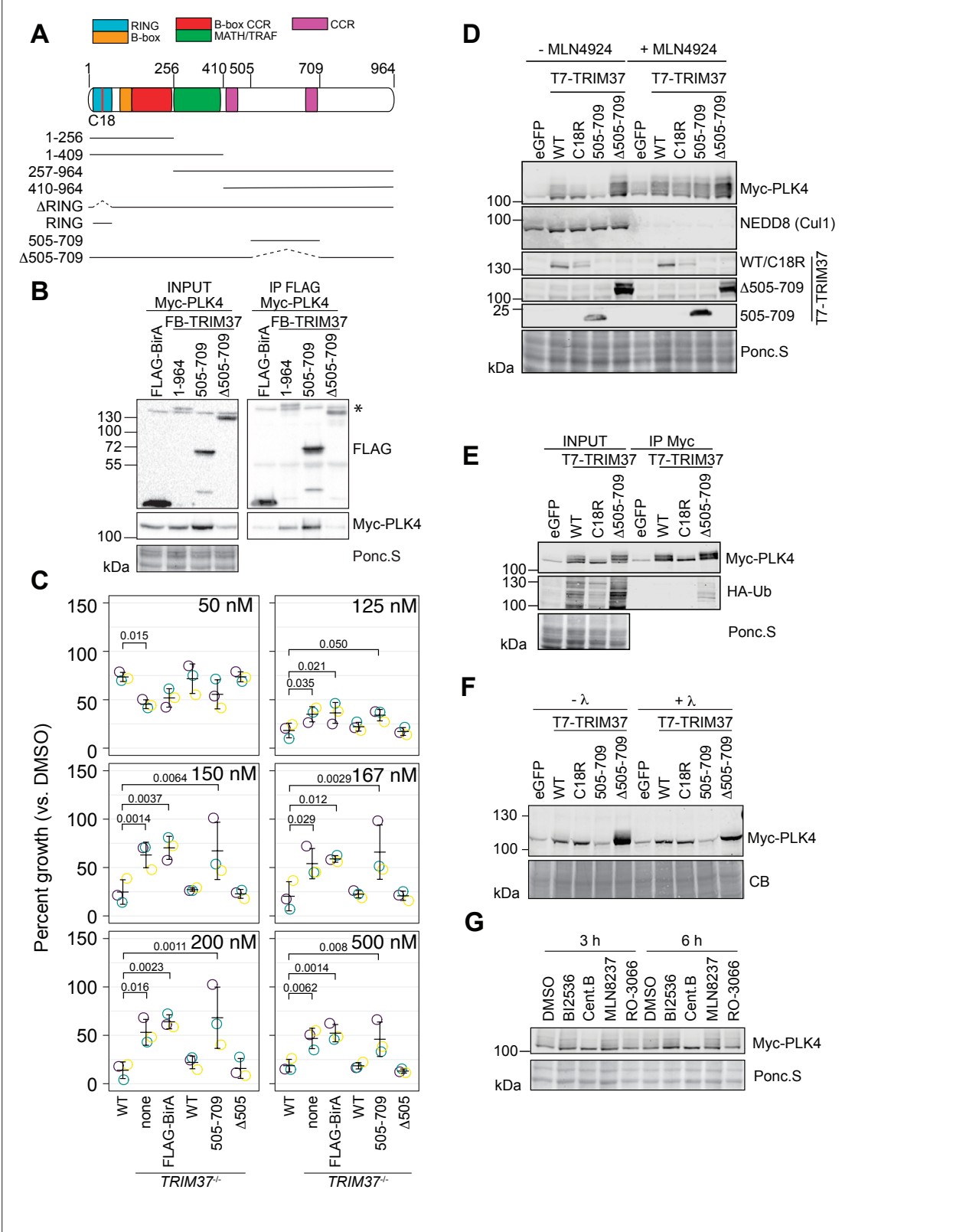

**Figure 7.** TRIM37 promotes PLK4 phosphorylation in an E3-dependent manner. (**A**) TRIM37 domain schematic. Constructs used for structure-function experiments indicated below. (**B**) RPE-1cells were transfected to express Myc-PLK4 and FLAG-BirA or the indicated FB-TRIM37 fusion protein (top). Cells were lysed and subjected to anti-FLAG immunoprecipitation. Input and immunoprecipitates were analyzed by immunoblotting for the FLAG-BirA fusions (FLAG) or for Myc-PLK4. Ponc.S indicates total protein. * indicates position of FLAG-Cas9. (**C**) WT RPE-1, *TRIM37*⁻/⁻ and *TRIM37*⁻/⁻ expressing FB or

*Figure 7 continued on next page*

*Figure 7 continued*

the indicated FB-TRIM37 construct were seeded for clonogenic assays and grown in DMSO or the indicated concentration of centrinone B for 14 days. Colony density was quantified and growth compared to that in DMSO determined. Means from each replicate are shown as open circles. Resulting mean and standard deviation shown (n=3). Significant *p*-values (< 0.05) from Dunnett post hoc test using 'WT' as a control after one-way ANOVA shown. Note that the results from this experiment and those in *Figure 4B* are from the same experiment; therefore 'WT', '*TRIM37*⁻ᐟ⁻ none', '*TRIM37*⁻ᐟ⁻ FLAG-BirA'', and '*TRIM37*⁻ᐟ⁻ WT' are duplicated in these panels. (**D**) HEK293T cells transfected to express Myc-PLK4 and the indicated protein (top) were grown overnight and subsequently treated with DMSO or MLN4924 for 22 hr and MG132 for the final 4 hr. Cell extracts were prepared and probed by Western blot using the indicated antibodies. Ponc.S indicates total protein. (**E**) HEK293T cells were transfected to express Myc-PLK4, HA-Ub, and the indicated protein (top). Cells were harvested after 48 hr and subjected to immunoprecipitation using anti-Myc antibodies. Input and immunoprecipitates were analyzed by immunoblotting for PLK4 and HA-Ub. Ponc.S indicates total protein. (**F**) HEK293T cells were transfected to express Myc-PLK4 and eGFP or the indicated T7-TRIM37 protein (top) for 48 hr. MG132 was added for the final 4 hr. Lysates were mock treated (-λ) or incubated with λ-phosphatase (+λ) and subsequently subjected to immunoblot for PLK4. (**CB**) indicates total protein. (**G**) HEK293T cells were transfected to express Myc-PLK4 and T7-TRIM37 Δ505–709 for 48 hr. Cells were treated with the indicated inhibitor (top) for 3 or 6 hr and analyzed by immunoblot for PLK4. Ponc.S indicates total protein. See *Figure 7—source data 1*.

The online version of this article includes the following source data and figure supplement(s) for figure 7:

**Source data 1.** Source data for *Figure 7*.

**Figure supplement 1.** Characterization of TRIM37 interactions with PLK4.

**Figure supplement 1—source data 1.** Source data for *Figure 7—figure supplement 1*.

**Figure supplement 2.** Summary of TRIM37 rescue constructs used and resulting phenotypes.

phosphorylation (*Figure 7G*), we monitored GFP-PLK4 mobility by FRAP in WT, *TRIM37*⁻ᐟ⁻ cells and after TRIM37 siRNA but did not observe any differences compared to control cells indicating that the phospho-forms of PLK4 stabilized by TRIM37 likely lie outside the phosphodegron region (*Figure 7—figure supplement 1H, I*). Together these data suggest that TRIM37 promotes the accumulation of phosphorylated PLK4 in an E3-dependent manner but this phenomenon does not require robust interaction with PLK4 itself.

## Discussion

Most animal cells harboring abnormal centrosome numbers are subject to p53-dependent growth arrest and the mechanisms of these pathways are beginning to be understood. Here, by leveraging the various phenotypes caused by treatment of cells with different concentrations of the selective PLK4 inhibitor centrinone B, we uncover multiple pathways leading to growth arrest in response to abnormal centrosome numbers. Not only does centrosome number play a role, but we hypothesize that properties or the activity of PLK4 itself can also trigger growth arrest. Curiously, we found that TRIM37 is required for growth arrest in some, but not all centrinone B concentrations tested.

Several observations support our hypothesis that differential PLK4 inhibition used for our screens resulted in distinct cellular states. First, we observed clear differences in centrosome number. Second, MDM2 was cleaved using 200 nM, but not 500 nM centrinone B consistent with excess centrosomes in the former condition. Last, the genes derived from each screen were distinct. Specifically, we identified ANKRD26/PIDDosome only in the presence of excess centrosomes and also identified centriole proteins that, when disrupted, could decrease centrosome load, although we did not formally test this. We note that 200 nM centrinone B was not optimal to induce excess centrosomes in A375 cells (compare *Figure 4C* with *Figure 4—figure supplement 1*), yet we still identified genes that overlapped with those from the comparable RPE-1 screen suggesting that these cells were subjected to similar conditions.

Centrosome amplification after partial inhibition of PLK4 has been previously observed using CFI-400495 (*Mason et al., 2014*), YLT-11 (*Lei et al., 2018*), or analog-sensitive alleles of *PLK4* (*Moyer et al., 2015*). Current models suggest that partially inhibited PLK4 reduces its auto-phosphorylation required for its degradation. As a consequence, PLK4 accumulates and promotes centriole overduplication (*Holland and Cleveland, 2014*). While TRIM37 is involved in mediating growth arrest to partial PLK4 inhibition, it is not required for arrest after PLK4 overexpression, which also leads to extra centrosomes (*Evans et al., 2021*). The fundamental difference between centrosome amplification caused by PLK4 overexpression or by partial inhibition may relate to the per molecule activity of PLK4, perhaps pointing a role for TRIM37 in regulating PLK4 activity. Our FRAP data indicated a

dose-dependent decrease in phosphorylated, and therefore active, PLK4 upon increasing centrinone B that correlated well with growth arrest activity. If altered PLK4 activity, and not extra centrosomes, is responsible for growth arrest, why would we identify proteins such as ANKRD26 and the PIDDosome that have clear roles in response to supernumerary centrosomes (*Evans et al., 2021*; *Burigotto et al., 2021*; *Fava et al., 2017*)? Growth suppression after PLK4 inhibition at any concentration of centrinone B was only partially TRIM37-dependent, suggesting that multiple pathways might be activated in these conditions, one dependent on centrosomes and the other dependent on PLK4. Comparing our dataset with that of the PLK4 overexpression screen (*Evans et al., 2021*) yields an overlap of 22 genes that we propose are involved in a response to supernumerary centrosomes (*Figure 2—figure supplement 2G*). We suggest that the genes unique to our dataset (i.e., low-dose centrinone B treatment) might modulate the response to inhibited PLK4. *CEP85* and *USP9X* are such genes and both encode proteins that affect STIL, a regulator of PLK4 activity. CEP85 is required for robust STIL interaction with PLK4 while USP9X stabilizes STIL (*Liu et al., 2018*; *Kodani et al., 2019*). Interestingly, 53BP1 and USP28 were similarly dispensable for growth arrest after PLK4 overexpression (*Evans et al., 2021*). In this case, the MDM2 p60 fragment that is known to interact with and stabilize p53 (*Oliver et al., 2011*) might be redundant with the function of 53BP1-USP28. It is therefore an exciting possibility that PLK4 activity, by itself, may be a determinant of p53-dependent cell cycle arrest.

TRIM37 is proposed to mediate the degradation of CEP192 that in turn affects PCM assembly in mitotic cells treated with PLK4 inhibitors (*Yeow et al., 2020*; *Meitinger et al., 2020*), however, there is no consensus on the exact centrosome or centriole TRIM37 targets. We observed a decrease in overall CEP192 protein levels, but not those of PCNT, CEP215, or CEP120 after stable expression of FB-TRIM37 (*Figure 6B*). The effect of TRIM37 expression on centrosomal proteostasis is unclear. Two previous studies examining centrosomal protein abundance after TRIM37 expression detected decreases in overall CEP192, however one report did not detect changes in CEP215 or CEP152 (*Meitinger et al., 2020*), while another detected decreases in CEP215 and PCNT (*Yeow et al., 2020*). We did not detect reciprocal changes in protein levels in *TRIM37$^{-/-}$* cells (*Figure 6A*) suggesting that the effects on PCM are dependent on overexpressed TRIM37 or that the *TRIM37$^{-/-}$* cell line acquired a genetic or epigenetic change that suppresses effects on PCM proteins. TRIM37 overexpression also decreased the amount of CEP192, PCNT, and CEP120 detected at mitotic centrosomes (*Figure 6D* and *Figure 6—figure supplement 1D*). More work will be needed to identify direct and indirect targets of TRIM37 at the centrosome.

The E3 ligase activity of TRIM37 was required for changes in the bulk abundance of CEP192 and for the reduction in CEP192, PCNT, and CEP120 proteins at mitotic centrosomes. In contrast, we observed dosage-dependent phenotypes for the E3 ligase mutant TRIM37 C18R. Stable and high expression of this variant caused a strong growth arrest phenotype (*Figure 4A*) while lower expression using an inducible system resulted in partial growth arrest activity in response to centrinone B treatment (*Figure 4E and F*). To explain this phenomenon, we consider that either the expression of the C18R mutant binds to and sequesters TRIM37 targets to phenocopy the effect of degradation, or that higher levels of TRIM37 C18R drive the formation of an E3-independent complex important for growth arrest, as has been suggested for a TRIM37 ligase-independent role in autophagy (*Wang et al., 2018*).

Current models of TRIM37 growth arrest function after PLK4 inhibition have primarily focused on mitotic length (*Lambrus et al., 2016*; *Meitinger et al., 2016*; *Yeow et al., 2020*; *Meitinger et al., 2020*). After complete PLK4 inhibition, cells containing one or no centrosome arrested, but in these studies, mitotic length and growth arrest were not well correlated and not long enough to activate a 'mitotic timer' (*Wong et al., 2015*; *Lambrus et al., 2015*). This led to the speculation that small increases in mitotic length over multiple cell cycles might be equivalent to a single mitosis with a larger delay (*Lambrus et al., 2016*; *Meitinger et al., 2016*) but this possibility has not yet been tested. Here, however, we provide multiple lines of evidence that mitotic length may not be critical for growth arrest after PLK4 inhibition. First, in WT RPE-1 cells, we observed that treatment with 50 nM and 125 nM centrinone B led to ~25% and 90% decreases in cell proliferation, respectively (*Figure 4B*) without causing concomitant increases in mitotic length (*Figure 5A*). Second, when comparing WT and *TRIM37$^{-/-}$* cells, we observed a significant difference in growth between these cell lines at 125 nM centrinone B or greater but did not detect significant differences in mitotic length until a treatment with 200 nM centrinone B. Third, induced expression of TRIM37 C18R caused a partial reduction in

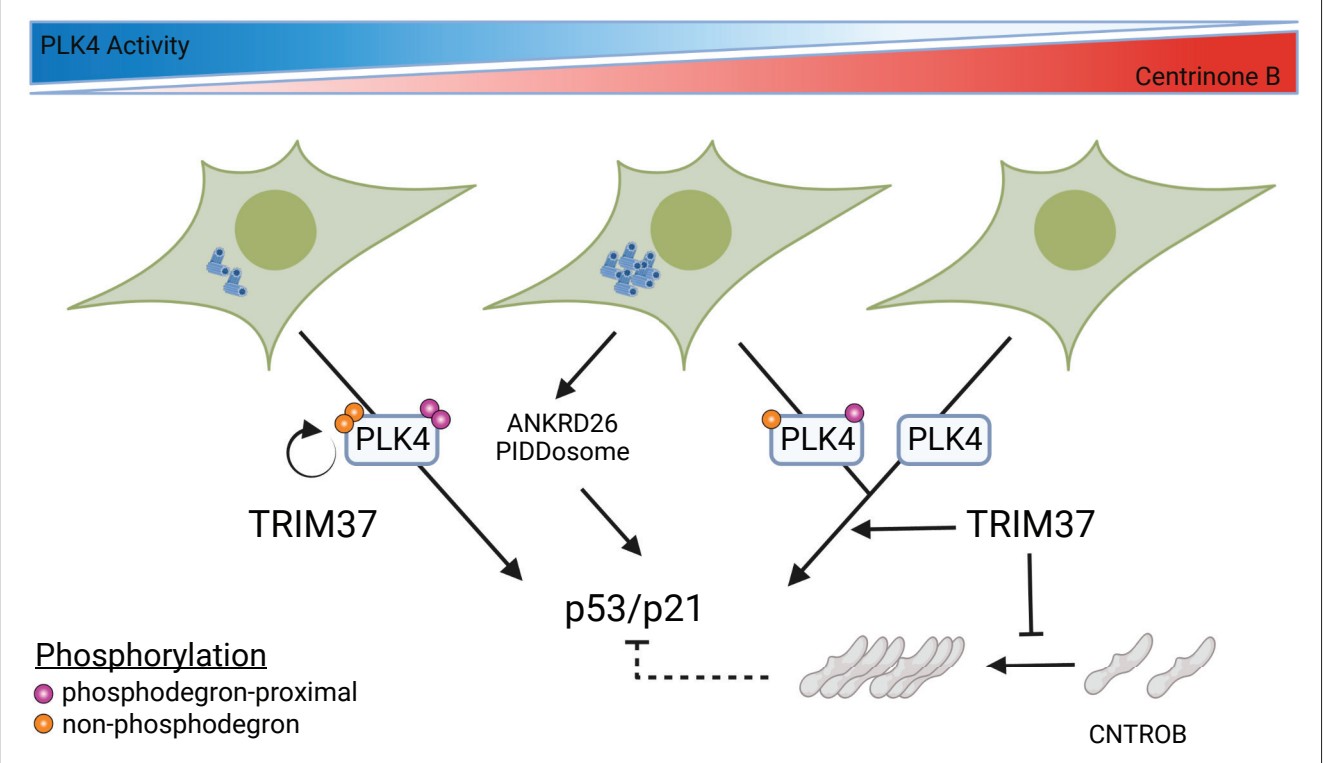

**Figure 8.** Model for growth arrest and TRIM37 growth arrest activity. PLK4 activity decreases in a dose-dependent manner upon centrinone B addition. TRIM37 promotes PLK4 auto-phosphorylation (orange circles) outside the phosphodegron region (purple circles). PLK4 inhibition initially results in TRIM37-independent growth arrest. Continued addition of centrinone B results in centrosome overduplication that is detected by the ANKRD26/ PIDDosome pathway in addition to a TRIM37-dependent growth arrest pathway. Complete inhibition of PLK4 results in TRIM37-dependent growth arrest. TRIM37 also prevents the appearance of CNTROB-dependent aggregates. We hypothesize that these aggregates might affect p53/p21 activation (dotted lines) (created with https://biorender.com/).

cell proliferation without any apparent effect on mitotic length (*Figure 4E and F*). We also note that PLK4 overexpression in mice or flies results in increased mitotic indices (*Marthiens et al., 2013*; *Basto et al., 2008*) indicative of lengthened mitoses, yet TRIM37 is not required for growth arrest under similar conditions in cultured cells. Mitotic length after treatment with 200 and 500 nM centrinone B is clearly affected by TRIM37 but our data does not suggest that this directly influences growth arrest.

Although others have observed TRIM37-dependent ubiquitination of PLK4 (*Meitinger et al., 2020*), we detected low amounts of ubiquitinated PLK4 only after co-expression with the TRIM37Δ505–709 mutant (*Figure 7E*). In contrast, we found that TRIM37 primarily promoted PLK4 phosphorylation in a manner that was dependent on PLK4 activity itself (*Figure 7F and G*). We did not find that PLK4 mobility by FRAP was affected by the loss of TRIM37 in the absence or presence of centrinone B (*Figure 7—figure supplement 1H, I*) suggesting that the phosphorylation sites stabilized by TRIM37 lie outside the PLK4 phosphodegron-adjacent regions monitored by this method (*Yamamoto and Kitagawa, 2019*). It will be of interest to determine how TRIM37 promotes PLK4 phosphorylation, which regions of PLK4 are modified, and to assess if these phosphorylation events are contributing to TRIM37-dependent growth arrest in response to abnormal centrosome numbers or altered PLK4 activity.

We described three distinct growth arrest phases after PLK4 inhibition characterized by cellular centrosome number abnormalities (*Figure 4E*). Our FRAP assays indicated that the mobility of PLK4 decreased after centrinone B treatment in a dose-dependent manner that mirrored the growth arrest activity of WT cells. We therefore propose that a direct aspect of PLK4 activity, either PLK4 itself or a substrate of PLK4, underlies the growth arrest after PLK4 inhibition (*Figure 8*). A level of PLK4 inhibition that does not affect centriole number initiates p53 arrest independent of TRIM37, but cell arrest after further PLK4 inhibition becomes dependent on TRIM37. TRIM37 itself promotes phosphorylation

of PLK4 and can affect the abundance and/or localization of CEP192, PCNT, and CEP120, but the effect of these functions is not clear. SASS6 was required for growth arrest at all centrinone B concentrations, (*Figure 4—figure supplement 3*) suggesting that centrioles themselves may play a role to integrate the growth arrest signal.

In closing, we used CRISPR/Cas9 screening to characterize the global, dose-dependent response to PLK4 inhibition. While previous studies focused on complete PLK4 inactivation and mitotic length after centrosome loss as a potential mechanism driving p53-dependent cell cycle arrest, we found that the loss of PLK4 activity better correlates with the subsequent growth arrest. Proteomic analysis of PLK4 substrates and the cellular aggregates that form in the absence of TRIM37 will be required to provide mechanistic details of this pathway and may yield the identification of PLK4 substrates underpinning this response.

# Materials and methods

## Key resources table

| Reagent type (species) or resource | Designation | Source or reference | Identifiers | Additional information |
|---|---|---|---|---|
| Cell line (human) | RPE-1, Epithelial (female, immortalized with hTERT) | ATCC | CRL-4000 | |
| Cell line (human) | A375, Epithelial (female, malignant melanoma) | ATCC | CRL-1619 | |
| Cell line (human) | RPE-1 Cas9 | *Zimmermann et al., 2018* | | |
| Cell line (human) | A375 Cas9 | *Hart et al., 2015* | | |
| Cell line (human) | HEK 293T, Epithelial (female, fetal kidney) | ATCC | CRL-3216 | |
| Cell line (human) | RPE-1 $TRIM37^{-/-}$ (clone) | This study | | Created by transfecting RPE-1 Cas9 with sgRNA TRIM37 1. Single clones selected and screened for TRIM37 disruption by PCR and Western blot. |
| Cell line (humanl) | RPE-1 $TRIM37^{-/-}$ (pool) | This study | | Created by transfecting RPE-1 Cas9 with sgRNA TRIM37 e5. Pools selected by treatment with centrinone B. |
| Cell line (human) | A375 $TRIM37^{-/-}$ (pool) | This study | | Created by transfecting A375 Cas9 with sgRNA TRIM37 e5. Pools selected by treatment with centrinone B. |
| Recombinant DNA reagent (plasmid, viral library) | TKOv1 library | *Hart et al., 2015* | | |
| Recombinant DNA reagent (plasmid) | plentiGuide-Puro | *Sanjana et al., 2014* | | |
| Recombinant DNA reagent (plasmid) | pLgP TRIM37sg1 | This study | | Cloning details in Materials and methods |
| Recombinant DNA reagent (plasmid) | pLgP TRIM37sg2 | This study | | Cloning details in Materials and methods |
| Recombinant DNA reagent (plasmid) | pcDNA5-FRT/TO-Myc-PLK4 | This study | | Cloning details in Materials and methods |
| Recombinant DNA reagent (plasmid) | pcDNA5 FLAG-BirA | *Gupta et al., 2015* | | Cloning details in Materials and methods |
| Recombinant DNA reagent (plasmid) | pcDNA5 FLAG-BIrA-TRIM37 | *Gupta et al., 2015* | | Cloning details in Materials and methods |
| Recombinant DNA reagent (plasmid) | pcDNA5 FLAG-BirA-TRIM37 C18R | This study | | Cloning details in Materials and methods |

*Continued on next page*

*Continued*

| Reagent type (species) or resource | Designation | Source or reference | Identifiers | Additional information |
|---|---|---|---|---|
| Recombinant DNA reagent (plasmid) | pcDNA5 FLAG-BirA-TRIM37 RING | This study | | Cloning details in Materials and methods |
| Recombinant DNA reagent (plasmid) | pcDNA5 FLAG-BirA-TRIM37 ΔRING | This study | | Cloning details in Materials and methods |
| Recombinant DNA reagent (plasmid) | pcDNA5 FLAG-BirA-TRIM37 1–256 | This study | | Cloning details in Materials and methods |
| Recombinant DNA reagent (plasmid) | pcDNA5 FLAG-BirA-TRIM37 257–964 | This study | | Cloning details in Materials and methods |
| Recombinant DNA reagent (plasmid) | pcDNA5 FLAG-BirA-TRIM37 1–409 | This study | | Cloning details in Materials and methods |
| Recombinant DNA reagent (plasmid) | pcDNA5 FLAG-BirA-TRIM37 410–964 | This study | | Cloning details in Materials and methods |
| Recombinant DNA reagent (plasmid) | pcDNA5 FLAG-BirA-TRIM37 505–709 | This study | | Cloning details in Materials and methods |
| Recombinant DNA reagent (plasmid) | pcDNA5 FLAG-BirA-TRIM37 Δ505–709 | This study | | Cloning details in Materials and methods |
| Recombinant DNA reagent (plasmid) | pSIN FLAG-BirA | This study | | Cloning details in Materials and methods |
| Recombinant DNA reagent (plasmid) | pSIN FLAG-BirA-TRIM37 | This study | | Cloning details in Materials and methods |
| Recombinant DNA reagent (plasmid) | pSIN FLAG-BIrA-TRIM37 C18R | This study | | Cloning details in Materials and methods |
| Recombinant DNA reagent (plasmid) | pSIN FLAG-BirA-TRIM37 ΔRING | This study | | Cloning details in Materials and methods |
| Recombinant DNA reagent (plasmid) | pSIN FLAG-BirA-TRIM37 505–709 | This study | | Cloning details in Materials and methods |
| Recombinant DNA reagent (plasmid) | pSIN FLAG-BirA-TRIM37 Δ505–709 | This study | | Cloning details in Materials and methods |
| Recombinant DNA reagent (plasmid) | pInduce PLK4 3xFLAG | This study | | Cloning details in Materials and methods |
| Recombinant DNA reagent (plasmid) | pcDNA3-HA-ubiquitin | This study | | |
| Recombinant DNA reagent (plasmid) | pcDNA5-FRT/TO-eGFP | *Kean et al., 2011* | | |
| Recombinant DNA reagent (plasmid) | p T7 TRIM37 | This study | | Cloning details in Materials and methods |

*Continued on next page*

*Continued*

| Reagent type (species) or resource | Designation | Source or reference | Identifiers | Additional information |
|---|---|---|---|---|
| Recombinant DNA reagent (plasmid) | p T7 TRIM37 C18R | This study | | Cloning details in Materials and methods |
| Recombinant DNA reagent (plasmid) | p T7 TRIM37 505–709 | This study | | Cloning details in Materials and methods |
| Recombinant DNA reagent (plasmid) | p T7 TRIM37 Δ505–709 | This study | | Cloning details in Materials and methods |
| Recombinant DNA reagent (plasmid) | pcDNA5 eGFP PLK4 | *Yamamoto and Kitagawa, 2019* | | |
| Recombinant DNA reagent (plasmid) | pcDNA5 eGFP PLK4 kinase +L1 | *Yamamoto and Kitagawa, 2019* | | |
| Sequence-based reagent | NGS outer FOR | *Hart et al., 2015* | | AGGGCCTATTTCCCATGATTCCTT |
| Sequence-based reagent | NGS outer REV | *Hart et al., 2015* | | TCAAAAAAGCACCGACTCGG |
| Sequence-based reagent | TRIM37 sgRNA 1 forward | This study | | CACCGACTTCAGGAGGTGGAGCACC |
| Sequence-based reagent | TRIM37 sgRNA 1 reverse | This study | | AAACGGTGCTCCACCTCCTGAAGTC |
| Sequence-based reagent | TRIM37 sgRNA 2 forward | This study | | CACCGTCGTAGCTGGAGTGGAGCAC |
| Sequence-based reagent | TRIM37 sgRNA 2 reverse | This study | | AAACGTGCTCCACTCCAGCTACGAC |
| Sequence-based reagent | TRIM37 sgRNA 1 IVT forward | This study | | GGATCCTAATACGACTCACTATAGGGACTTCAGGAGGTGGAGCACC |
| Sequence-based reagent | TRIM37 sgRNA 1 IVT reverse | This study | | TTCTAGCTCTAAAACGGTGCTCCACCTCCTGAAGTCCC |
| Sequence-based reagent | TRIM37 sgRNA 1 check forward | This study | | TCTGGCCCACTTTGTATTCTCT |
| Sequence-based reagent | TRIM37 sgRNA 1 check reverse | This study | | CCAGGTCAGGAGATCGAGAC |
| Sequence-based reagent | TRIM37 sgRNA exon 5 IVT forward | This study | | GGATCCTAATACGACTCACTATA GTCTGCCATCAGTGTGCACTT |
| Sequence-based reagent | TRIM37 sgRNA exon 5 IVT reverse | This study | | TTCTAGCTCTAAAACAAGTGCACACTGATGGCAGA |
| Sequence-based reagent | TRIM37 exon 5 check forward | This study | | AAGCACATGCCCAAAATGTAGT |
| Sequence-based reagent | TRIM37 exon 5 check reverse | This study | | GGGTCCATCAAACCACACAAAC |
| Sequence-based reagent | cr_tracr_RNA | This study | | GTTTTAGAGCTAGAAATAGCAAGTTAAAATAAGGCTAGTCCGTTATCAACTTGAAAAAGTGGCACCGAGTCGGGCTTTT |
| Sequence-based reagent | IVT forward | This study | | TAATACGACTCACTATAG |
| Sequence-based reagent | IVT reverse | This study | | AAAAGCACCGACTCGGTG |
| Sequence-based reagent | TRIM37 forward | This study | | ACTAGGCGCGCCAGATGAACAGAGCGTGGAG |

*Continued on next page*

*Continued*

| Reagent type (species) or resource | Designation | Source or reference | Identifiers | Additional information |
|---|---|---|---|---|
| Sequence-based reagent | TRIM37 reverse | This study | | TTAGGCGGCCGCTTACCTTCCACTATTTTCATCTGTATTG |
| Sequence-based reagent | TRIM37 256 reverse | This study | | TTAGGCGGCCGCTTACATGGGCTTCCGATGAACTTG |
| Sequence-based reagent | TRIM37 257 forward | This study | | ACTAGGCGCGCCAGCATCTTTTGTTACCACTCCTG |
| Sequence-based reagent | TRIM37 409 reverse | This study | | TTAGGCGGCCGCTTATTGAAAGAAAGTTGGTGAACGTAC |
| Sequence-based reagent | TRIM37 410 forward | This study | | ACTAGGCGCGCCAAAATCCCGGGACCAGCATTG |
| Sequence-based reagent | TRIM37 RING reverse | This study | | TTAGGCGGCCGCTTAATCAAGCTGTTGTGTTACTTCTTC |
| Sequence-based reagent | TRIM37 505 forward | This study | | ACTAGGCGCGCCACAGAATGAAGATTATCATCACGAGC |
| Sequence-based reagent | TRIM37 709 reverse | This study | | TTAGGCGGCCGCTTACATGTCTCCAGAAGCAGCAC |
| Sequence-based reagent | TRIM37 710 forward | This study | | ACTAGGCGCGCCACAGACAAGCCTTTTTTCTGCTG |
| Sequence-based reagent | TRIM37 Δ 505–709 forward | This study | | CAGACAAGCCTTTTTTCTG |
| Sequence-based reagent | TRIM37 Δ 505–709 reverse | This study | | AATCTTCTCCTCATCTTCTTC |
| Sequence-based reagent | TRIM37 C18R forward | This study | | TCCCGCAATTTCTCCATACGAATGAAACATCGGAAAACC |
| Sequence-based reagent | TRIM37 C18R reverse | This study | | GGTTTTCCGATGTTTCATTCGTATGGAGAAATTGCGGGA |
| Sequence-based reagent | TRIM37 Δ RING forward | This study | | GCTCCACTCCAGCTACGA |
| Sequence-based reagent | TRIM37 Δ RING reverse | This study | | TCGGAAAACCTCAGCAATG |
| Sequence-based reagent | Remove FLAG-BirA reverse | This study | | GGTACCAAGCTTAAGTTTAAAC |
| Sequence-based reagent | Remove FLAG-BirA forward | This study | | GGGGGATCTGGCCCCGGC |
| Sequence-based reagent | T7 tag forward | This study | | CAGCCTCCGGACTCTAGCGTTTAAACTTAAGCTTGGTACCATGG CCAGCATGACCGGCGGCCAGCAG |
| Sequence-based reagent | T7 tag reverse | This study | | CTCTGTTCATCTGGCGCGCCGCCGCCGGGGCCAGATCCCCCA CCCATCTGCTGGCCGCCGGTCATGCT |
| Sequence-based reagent | PLK4 for | This study | | TTGGCGCGCCAATGGCGACCTGCATCGGG |
| Sequence-based reagent | PLK4 rev | This study | | CCGCTCGAGTTAACATTCTTGTTGGATTATCTCA |
| Sequence-based reagent | CEP120 siRNA siGENOME | *Comartin et al., 2013* | | GAUGAGAACGGGUGUGUAU |
| Sequence-based reagent | TRIM37 siRNA ON-TARGETplus SMARTpool | This study, Dharmacon | | GGACUUUGCUGGAGGUUAA, AUACGAAACUCCACAAUA, AGAGUGAGUUGAUAUCUAA, GAAUGUAGAAGCUGUAAGA |
| Sequence-based reagent | Non-target #4 | Dharmacon | | AUGAACGUGAAUUGCUCAA |
| Sequence-based reagent | Luciferase GL2 control | Dharmacon | | CGUACGCGGAAUACUUCGA |

*Continued on next page*

*Continued*

| Reagent type (species) or resource | Designation | Source or reference | Identifiers | Additional information |
|---|---|---|---|---|
| Antibody | Anti-CEP135 (rabbit, polyclonal) | *Bird and Hyman, 2008* | | IF (1:1000) |
| Antibody | Anti-p53 (mouse, monoclonal) | Santa Cruz Biotechnology | sc-126 | Western blot (1:250) IF (1:250) |
| Antibody | p21 (mouse, monoclonal) | Santa Cruz Biotechnology | sc-817 | Western blot (1:200) IF (1:200) |
| Antibody | Mdm2 (mouse, monoclonal) | MilliporeSigma | MABE340 | Western blot (1:200) |
| Antibody | γ-Tubulin (mouse, monoclonal) | MilliporeSigma | T6557 | Western blot (1:1000) |
| Antibody | TRIM37 (rabbit, polyclonal) | Bethyl Laboratories | A301-174A | Western blot (1:250) IF (1:250) |
| Antibody | CEP120 (rat, polyclonal) | PMID:29741480 | | Western blot (1:1000) IF (1:4000) |
| Antibody | CETN2 (mouse, monoclonal) | MilliporeSigma | 04-1624 | IF (1:1000) |
| Antibody | FLAG (mouse, monoclonal) | MilliporeSigma | F7425 | Western blot (1:1000) IF (1:1000) |
| Antibody | PLK4 (mouse, monoclonal) | MilliporeSigma | MABC544 | Western blot (1:500) IF(1:250) |
| Antibody | BirA (mouse, monoclonal) | Novus Biologicals | NBP2-59939 | IF (1:1000) |
| Antibody | Centrobin (rabbit, polyclonal) | Proteintech | 26880-1-AP | IF (1:1000) |
| Antibody | CEP192 (rabbit, polyclonal) | Bethyl Laboratories | A302-324 | IF (1:1000) |
| Antibody | CEP192 (rabbit, polyclonal) | *Pelletier et al., 2004* | | Western blot (1:500) |
| Antibody | PCNT (rabbit, polyclonal) | Abcam | ab4448 | Western blot (1:500) IF (1:1000) |
| Antibody | PCNT (mouse, monoclonal) | Abcam | ab28144 | IF (1:1000) |
| Antibody | SASS6 (rabbit, polyclonal) | *Dammermann et al., 2004* | | Western blot (1:5000) |
| Antibody | SASS6 (goat, polyclonal) | Santa Cruz Biotechnology | sc-81431 | IF (1:300) |
| Antibody | Glutamylated tubulin (GT335) (mouse, monoclonal) | Adipogen | AG-20B-0020-C100 | IF (1:1000) |
| Antibody | CEP97 (goat, polyclonal) | Santa Cruz Biotechnology | sc-100028 | IF (1:250) |
| Antibody | CEP215 (rabbit, polyclonal) | MilliporeSigma | 06-1398 | Western blot (1:500) IF (1:1000) |
| Antibody | T7 (mouse, monoclonal) | MilliporeSigma | 69522-3 | Western blot (1:1000) |
| Antibody | HA (mouse, monoclonal) | Covance | MMS-101R | Western blot (1:500) |
| Antibody | Myc (goat, polyclonal) | Abcam | ab9132 | Immunoprecipitation (1 µg) |
| Antibody | Anti-mouse Alexa Fluor 488 (donkey, polyclonal) | Thermo Fisher Scientific | A21202 | IF (1:500) |
| Antibody | Anti-rabbit Alexa Fluor 568 (donkey, polyclonal) | Thermo Fisher Scientific | A10042 | IF (1:500) |
| Antibody | Anti-rat Alexa Fluor 647 (donkey, polyclonal) | Jackson ImmunoResearch Laboratories | 712-605-153 | IF (1:500) |
| Antibody | Anti-goat Alexa Fluor 647 (donkey, polyclonal) | Thermo Fisher Scientific | A21447 | IF (1:500) |
| Antibody | Anti-mouse HRP | Bio-Rad Laboratories | 170-6516 | Western blot (1:5000) |

*Continued on next page*

*Continued*

| Reagent type (species) or resource | Designation | Source or reference | Identifiers | Additional information |
|---|---|---|---|---|
| Antibody | Anti-rabbit HRP | Bio-Rad Laboratories | 170-6515 | Western blot (1:5000) |
| Antibody | Anti-rabbit IRDye 800CW | LI-COR | 926-32211 | Western blot (1:10,000) |
| Antibody | Anti-mouse IRDye 680RD | LI-COR | 926-8070 | Western blot (1:10,000) |
| Chemical compound, drug | DAPI | Invitrogen/Thermo Fisher Scientific | D21490 | 500 ng/mL |
| Chemical compound, drug | Prolong Gold antifade reagent | Life Technologies/Thermo Fisher Scientific | P36930 | |
| Chemical compound, drug | Centrinone B | Tocris Bioscience | 1384545 | Used as indicated |
| Chemical compound, drug | Nutlin-3a | Cayman Chemical | 10004372-1 | 600 nM |
| Chemical compound, drug | RO-3306 | Selleck Chemicals | S7747 | 10 mM |
| Chemical compound, drug | BI-2536 | ChemieTek | CT-BI2536 | 100 nM |
| Chemical compound, drug | MLN8237 | Selleck Chemicals | S1133 | 200 nM |
| Chemical compound, drug | MG132 | Selleck Chemicals | S2619 | 10 mM |
| Chemical compound, drug | G418 | WISENT Bioproducts | 400-130-IG | Used as indicated |
| Chemical compound, drug | SiR-DNA | Spirochrome | CY-SC007 | 200 nM |
| Software | SoftWoRx software | | RRID:SCR_019157 | |
| Software | CellProfiler Image Analysis Software | Broad Institute | RRID:SCR_007358 | |
| Software | R Project for Statistical Computing | | RRID:SCR_001905 | |
| Software | Fiji | Max Planck Institute of Molecular and Cell Biology and Genetics; Dresden; Germany | RRID:SCR_002285 | |
| Software | NIS-Elements | | RRID:SCR_014329 | |
| Software | LI-COR Image Studio Software | | RRID:SCR_015795 | |
| Commercial assay or kit | HiScribe T7 High Yield RNA Synthesis Kit | New England Biolabs | E2040S | |
| Commercial assay or kit | Agencourt RNAClean XP | Beckman Coulter | A63987 | |
| Commercial assay or kit | QIAamp DNA Blood Maxi Kit | Qiagen | 51194 | |
| Commercial assay or kit | QIAprep Spin Miniprep Kit | Qiagen | 27106 | |
| Commercial assay or kit | Lipofectamine RNAiMAX | Life Technologies/Thermo Fisher Scientific | 13778-150 | |
| Commercial assay or kit | Lipofectamine 3000 Transfection Reagent | Life Technologies/Thermo Fisher Scientific | L3000015 | |
| Commercial assay or kit | KAPA HiFi HotStart ReadyMix | Kapa Biosystems | KK2601 | |

*Continued on next page*

*Continued*

| Reagent type (species) or resource | Designation | Source or reference | Identifiers | Additional information |
|---|---|---|---|---|
| Commercial assay or kit | Q5 Site-Directed Mutagenesis Kit | New England Biolabs | E0554S | |
| Commercial assay or kit | Gibson Assembly Master Mix | New England Biolabs | E2611 | |
| Commercial assay or kit | QuikChange Multi Site Directed Mutagenesis Kit | Agilent | 200513 | |

## Cell culture and drug treatments

All cell lines were cultured in a 5% $CO_2$ humidified atmosphere at 37°C. HEK293T (female, human embryonic kidney epithelial), hTERT RPE-1 (female, human epithelial cells immortalized with hTERT), and A375 cells (female, human malignant melanoma epithelial) are from ATCC. hTERT RPE-1 and A375 stably expressing Cas9 were from D Durocher (*Hart et al., 2015*; *Zimmermann et al., 2018*). All references to RPE-1 and A375 cells herein refer to hTERT RPE-1 or A375 stably expressing Cas9. RPE-1, HEK293T, and A375 were grown in Dulbecco's modified Eagle's medium (Gibco) supplemented with 10% (v/v) fetal bovine serum (FBS; Gibco) and 2 mM Glutamax (Gibco). PLK4 inhibitor centrinone B (Tocris) was used as described. Nutlin-3a (Cayman Chemical) was used at 600 nM. The CDK1, PLK1, and Aurora A kinase inhibitors RO-3306 (Selleck Chemicals), BI-2536 (ChemieTek), and MLN8237 (Selleck Chemicals) were used at 10 µM, 100 nM, and 200 nM, respectively. MG132 (Selleck Chemicals) was used at 10 µM. G418 (WISENT Bioproducts) was used at 600 µg/mL for cell selection and 200 µg/mL for routine culture. All cell lines used have been authenticated by STR profiling and tested negative for mycoplasma contamination.

## Plasmid construction

TRIM37 C18R was created with pcDNA5 FB TRIM37 as a template using site-directed mutagenesis (QuikChange, Agilent). Truncation mutants were created by PCR using pcDNA FB TRIM37 as a template and ligated into pcDNA5 FLAG-BirA digested with NotI and AscI. Internal deletions were created using pcDNA5 FB TRIM37 as a template using the Q5 Site-Directed Mutagenesis kit (NEB). pSIN constructs were created by amplifying the insert from the corresponding pcDNA5 plasmid and using Gibson cloning (NEB) to ligate into pSIN previously digested with BamHI and NotI. To create T7-tagged TRIM37, first FLAG-BirA was removed from pcDNA5 Flag-BirA by PCR. The appropriate TRIM37 mutant was amplified with primers encoding the T7 tag and inserted in the pcDNA5 template using Gibson cloning. PLK4 was amplified from cDNA and ligated into pcDNA5-FRT/TO-Myc using AscI and XhoI. sgRNA guide sequences were cloned into pLentiguidePuro as described (*Sanjana et al., 2014*).

## Virus production

To produce lentivirus, $4 \times 10^6$ HEK293T were seeded in a T-75 flask and subsequently transfected with 4 µg of the appropriate transfer vector, 3 µg psPAX2, and 2 µg pCMV-VSV-G using 18 µL each Lipofectamine 3000/P3000 reagent (Invitrogen). After 24 hr, growth medium was replaced with fresh medium containing 30% FBS and viral supernatant was collected after a further 48 hr. Virus was stored at –80°C.

## CRISPR/Cas9 screening

CRISPR screens were performed as described (*Hart et al., 2015*; *Zimmermann et al., 2018*). Briefly, Cas9-expressing cells were transduced with the TKOv1 viral library (~90 k sgRNA) (*Hart et al., 2015*) at low MOI (~0.3) in the presence of 4 µg/mL polybrene. RPE-1 cells were selected as described (*Olivieri and Durocher, 2021*). A375 cells were selected using 2 µg/mL puromycin. 10× $10^6$ cells were harvested 4 days post-transduction and represents day 0. Cells were grown for 6 days before being split for drug treatment in technical triplicate and further grown for 21 days. A library coverage of >100 cells/sgRNA was maintained at each step. gDNA from cell pellets was isolated using a QIAamp Blood Maxi Kit (Qiagen) and genome-integrated sgRNA sequences were amplified using the KAPA HiFi HotStart ReadyMix (Kapa Biosystems). Sequencing libraries were made by addition of i5

and i7 multiplexing barcodes in a second round of PCR and the product gel purified using QIAquick Gel Purification kit (Qiagen). Libraries were sequenced using Illumina HiSeq2500 or NextSeq500. Sequence data was analyzed using MAGeCK (*Li et al., 2014*) to determine sgRNA distribution among the samples. Drug-treated samples at 21 days post-drug addition were compared to DMSO-treated cells at 12 days post-drug addition to equalize the number of cell doublings. Genes with FDR <0.05 were used for further analysis. The significant gene list for the RPE-1 200 nM screen is the union from two independent biological replicates.

## Network analysis and gene enrichment

High-scoring genes from MAGeCK analysis were visualized using Cytoscape (*Shannon et al., 2003*). General node arrangement was performed using the yFiles Organic Layout and manually modified to facilitate visualization. Each screen condition (200 nM centrinone B, 500 nM centrinone B, and Nutlin-3a) was considered as a source node, corresponding hits as target nodes, and FDR as edge attributes. Genes from the indicated datasets were analyzed using the ClueGo app within Cytoscape (*Bindea et al., 2009*). Enrichments for Biological Function (circles) or Cellular Component (hexagons) based on all experimental evidence was determined. Only pathways with p-value < 0.05 are shown. Nodes arranged using the yFiles Organic Layout.

## CRISPR/Cas9 gene disruption

For lentivirus-mediated gene disruption of *TRIM37*, sgRNA sequences were cloned into plentiGuide-Puro as described (*Sanjana et al., 2014*). RPE-1 Cas9 cells were infected with lentiviral particles and selected as described above for CRISPR/Cas9 screening. A Clonal *TRIM37*$^{-/-}$ line was generated using in vitro transcribed (IVT) sgRNA. IVT templates were created by PCR using cr_tracrRNA, IVT forward, IVT reverse, and sgRNA-specific oligonucleotides (TRIM37 sgRNA 1). PCR products were used directly as templates for IVT using HiScribe T7 transcription kit (NEB). Resulting RNA was purified using RNAClean XP beads (Beckman Coulter) and used to transfect RPE-1 cells using RNAiMAX (ThermoFisher) according to the manufacturer's instructions. Clonal lines were generated by limiting dilution and assessed for gene disruption by Western blot and TIDE (*Brinkman et al., 2014*) or Synthego ICE (Synthego Performance Analysis, ICE Analysis. 2019. v2.0. Synthego; accessed 9/19/2018) analyses. *TRIM37*$^{-/-}$ pools in RPE-1 and A375 cells were generated similarly using an sgRNA targeting exon 5. After transfection, cells were grown in medium containing 500 nM centrinone B for 2 weeks to select for *TRIM37* disruption before growth in normal medium.

## Stable cell line generation

To generate cell lines, 200,000 cells were seeded with serial aliquots of viral supernatant and 4 µg/mL polybrene (MilliporeSigma) in one well of a six-well plate. Medium was changed after 24 hr and appropriate drug selection was added after an additional 24 hr where required. For stable expression of FLAG-BirA rescue constructs, immunofluorescence was performed to ensure all cells expressed the appropriate transgene. Doxycycline-inducible lines were selected with 600 µg/mL G418 until control cells died. We used pools that showed approximately 30% survival after initial selection.

## siRNA conditions

For siRNA knockdown experiments, 200 k cells were seeded per well of a six-well plate. Cells were reverse transfected using the indicated siRNA trigger (Horizon Discovery, Dharmacon; *Supplementary file 3*). For each well, 5 µL of 20 µM siRNA was combined with 3 µL Lipofectamine RNAiMAX (ThermoFisher) in 125 µL OPTIMEM medium (Gibco). Media was replaced after 24 hr and cells processed after 72 hr.

## Immunofluorescence staining and microscopy

Cells were grown as indicated on No. 1.5 coverslips, washed once with PBS, and fixed with –20°C methanol for at least 10 min. All subsequent steps performed at room temperature (RT). Coverslips were rinsed with PBS and blocked with antibody solution (PBS, 0.5% (w/v) BSA and 0.05% Tween-20) for 15–30 min. Samples were incubated with primary antibodies (*Supplementary file 3*) for 1 hr, washed 3 × 5 min and incubated with secondary antibodies (*Supplementary file 3*) and DAPI (0.1 µg/mL) for 45 min. Coverslips were washed 3 × 5 min and mounted on slides using Prolong Gold

(Invitrogen). Deconvolution wide-field microscopy was performed using the DeltaVision Elite system equipped with an NA 1.42 60× PlanApo objective (Olympus) and an sCMOS 2048 × 2048 camera (Leica Microsystems). Each field was acquired with a z-step of 0.2 μm through the entire cell and deconvolved using softWoRx (v6.0, Leica Microsystems). Maximum intensity projections are shown (0.1080 μm/pixel). Display levels are the same for all images in a panel unless otherwise indicated.

For live imaging, 15,000 cells were seeded per well in an eight-well Lab-Tek II chamber slide. The next day fresh medium containing drug was added and cells incubated for 3 days. Fresh medium containing indicated drug and 200 nM SiR-DNA (Spirochrome) was added for 2 hr before imaging. Microscopy was performed using the DeltaVision Elite system equipped with an NA 0.75 U Plan S-Apo objective (Olympus) and an sCMOS 2048 × 2048 camera. Each field was acquired with 6 × 2 μm z-step every 5 min for 24 hr. The time between NEBD and full chromosome separation judged by nuclear morphology was quantified.

Super-resolution microscopy was performed on a three-dimensional structured-illumination microscope (3D-SIM) (OMX Blaze v4, Leica Microsystems) as described (*Mojarad et al., 2017*).

## Image analysis

All automated quantification pipelines were created using CellProfiler 3.0 (*McQuin et al., 2018*) (http://www.cellprofiler.org/).

*Figure 1A and B*: Nuclei were detected using the DAPI channel and objects subsequently used as a mask to measure intensity in p53 or p21 channels. An arbitrary cut-off based on the distribution of p21 or p53 intensities in untreated cells was used to score positive cells.

*Figure 3J*, *Figure 3—figure supplement 1*, *Figure 4—figure supplement 2*: A centrosomal (γ-tubulin) or centriole marker (CEP120) were used to define centrosome regions. The TRIM37 images were masked by the centrosome objects and total intensity was measured.

*Figure 3—figure supplement 2* and *Figure 7—figure supplement 1D*: Nuclei were detected using the DAPI channel. The nuclear objects were expanded and a ring surrounding each nucleus was used as a mask to measure the total intensity in the BirA channel. The mean and standard deviation of the measured intensities of control cells was determined and a cut-off of the mean + 2.5× the standard deviation was used to score positive cells.

*Figure 6D* and *Figure 6—figure supplement 1D*: Each image was manually cropped to include a single mitotic cell. Each channel was background subtracted using the lower quartile intensity of the entire image and each channel was segmented into objects using a robust background thresholding and the integrated intensity of each object was measured.

## Western blot

Cells were grown as indicated, washed once with PBS and resuspended directly in 2× SDS-PAGE sample buffer containing Benzonase (0.25 U/μL, MilliporeSigma) and heated at 95°C for 5 min. Proteins (typically 10–20 μg) were separated by SDS-PAGE and transferred to PVDF using a wet transfer apparatus (Bio-Rad). Total protein was detected by staining with PonceauS (MilliporeSigma) and scanning. All steps performed at RT unless indicated. Blocking and primary antibody incubations were performed using TBS-T (TBS + 0.05% Tween-20) with 0.5% skim-milk powder (Bioshop). Membranes were blocked for 30 min and incubated with primary antibody (*Supplementary file 3*) overnight at 4°C. After washing 3 × 5 min, membranes were incubated with secondary antibodies for 45. HRP-conjugated secondary antibodies (Bio-Rad) were incubated in TBS-T/milk for 45 min and washed 3 × 5 min with TBS-T before detecting using a Chemidoc imager (Bio-Rad). NearIR-conjugated secondary antibodies (LI-COR Biosciences) were incubated in TBS-T/milk + 0.015% SDS for 45, washed 3 × 5 min with TBS-T and 1 × 5 min with TBS before drying the membrane for 2 hr at RT. Dried blots were imaged using an Odyssey CLx imager (LI-COR Biosciences).

Quantification of Western blots were performed on images obtained using NearIR secondary antibodies. Images were quantified using Image Studio software (LI-COR Biosciences) and normalized to Ponceau S or Coomassie Blue staining of the same lane.

## Clonogenic survival assays

Two-hundred and fifty RPE-1 or 200 A375 cells were seeded in either a 10 cm dish or six-well plate. The next day medium was removed and medium containing the indicated drug was added. For

experiments using doxycycline-inducible cell lines, the media was refreshed every 3–4 days to ensure continued expression of the induced proteins. After 12–14 days, plates were rinsed once with PBS and fixed and stained with 0.5% crystal violet (MilliporeSigma) in 20% methanol for at least 20 min. Plates were washed extensively with water, dried, and scanned. Images were segmented using the Trainable Weka Segmentation tool (*Arganda-Carreras et al., 2017*) in ImageJ. A new model was built for each replicate if required. The resulting segmentation image was thresholded and used as a mask to overlay the original image that was inverted and background subtracted using a 50-pixel rolling circle, or the average of a region not containing colonies. The colony intensity per well or dish was then measured within the masked region.

## Immunoprecipitation and protein treatments

To detect complex formation between PLK4 and TRIM37, $2 \times 10^6$ RPE-1 cells were seeded per 10 cm dish and transfected with 3.75 μg Myc-PLK4 and 3.75 μg pcDNA5 FLAG-BirA construct using 15 μL Lipofectamine 3000/P3000 (ThermoFisher). Cells were harvested 24 hr post-transfection, washed once with PBS, and resuspended in lysis buffer (50 mM HEPES pH 8; 100 mM KCl; 2 mM EDTA; 10% glycerol; 0.1% NP-40; 1 mM DTT; protease inhibitors [Roche] phosphatase inhibitor cocktail 3 [MilliporeSigma]) for 30 min on ice. Lysates were frozen in dry ice for 5 min, then thawed and centrifuged for 20 min at $16,000 \times g$ at 4°C. An aliquot representing the input was removed before cleared supernatants were incubated with equilibrated anti-FLAG M2 Affinity Gel (MilliporeSigma) for 1–2 hr at 4°C. Beads were washed three times with lysis buffer before resuspension in 2× SDS-PAGE sample buffer. Samples were heated at 95°C for 5 min.

To probe PLK4 modification in *Figure 7D,F,G*, 350 k HEK293T cells were seeded per well of a six-well plate and subsequently transfected with 0.67 μg Myc-PLK4 and 1 μg T7-TRIM37 construct using 3.34 μL Lipofectamine 3000/P3000. Medium was changed after 6 hr and cells incubated for 48 hr in total before sampling. For *Figure 7G*, the indicated drug was added 3 and 6 hr before collection. Cells were collected directly in 2× SDS-PAGE sample buffer for *Figure 7D and G*. For *Figure 7F*, cells were collected and washed once with PBS. Cells were resuspended in a modified TNTE buffer (10 mM Tris-HCl, pH = 7.4, 100 mM NaCl, 1 mM EDTA, 1 mM DTT, 0.1% TX-100; protease inhibitors; ±phosphatase inhibitor cocktail 3) and incubated for 30 min on ice before addition of $MnCl_2$ and $\lambda$-phosphatase (Bio-Rad) to the appropriate samples for 30 min at 30°C. The soluble fractions were obtained by centrifugation at $16,000 \times g$ for 30 min. To probe for PLK4 ubiquitination, $1.5 \times 10^6$ HEK293T cells were seeded in a 10 cm dish and transfected with 2 μg Myc-PLK4, 2 μg HA-ubiquitin, and 2 μg eGFP or T7-TRIM37 with 12 μL Lipofectamine 3000/P3000. The medium was changed after 16 hr and cells harvested after 48 hr. Cells were washed once with PBS and resuspended in modified TNTE buffer and the soluble fraction was obtained as described above. Lysates were incubated with 3 μg anti-Myc antibodies (*Supplementary file 3*) and incubated for 1 hr at 4°C. Equilibrated Protein G Sepharose 4 Fast Flow beads (Cytiva) were added and samples further incubated for 1 hr at 4°C. Immunoprecipitates were washed 3× with modified TNTE buffer and eluted by addition of 2× SDS-PAGE sample buffer and heating at 95°C for 5 min.

## FRAP analysis

For experiments using disruption lines, 62.5 k cells were seeded per well of an eight-well LabTekII chamber. Cells were transfected with 400 ng pcDNA5 GFP-PLK4 or pcDNA5 GFP-PLK4 kin + L1 using 0.8 and 0.6 μL P3000/Lipofectamine 3000. Media was removed after 6 hr and replaced with media containing DMSO or centrinone B. Cells were incubated approximately 16 hr before imaging using a Nikon A1R-HD25 scanning laser confocal microscope with a LUN4 laser unit and GaAsP PMT. A single Z-slice was imaged in Galvano mode, 1.2 μs dwell time using a 488 nm excitation wavelength, a 521/42 bandpass emission filter, and a 60× NA 1.2 water immersion objective. GFP-PLK4 condensates were imaged at 0, 4, and 8 s before bleaching for 500 ms using 60% 488 laser power and 16 fps scan speed. Images were acquired every 4 s for a total of 90 s after bleaching. Imaging parameters were adjusted as needed between replicates (typically 1.2% laser power with gain setting of 10). Analysis was performed using NIS-elements 'time measurement' module. For each image an ROI was drawn around the targeted area, a similar unbleached area, and a background region. Where appropriate, the ROI was moved to track the structure of interest. The signal from the targeted area (ROI1) was background subtracted (ROI3) and then normalized using the unbleached area (ROI2) to correct for

photobleaching during imaging. Ten to 15 GFP-PLK4 condensates from different cells were analyzed per condition per replicate.

## Acknowledgements

We thank members of the Pelletier Lab for scientific feedback during the preparation of this work. We are grateful to Sally Cheung and Suzanna Prosser for proofreading the manuscript. The rabbit and rat CEP120 antibodies and CEP135 antibody were generous gifts from Li-Heui Tsai, Moe Mahjoub, and Alex Bird, respectively. The TKOv1 virus library was generously provided by Jason Moffat. The pcDNA3 HA-ubiquitin plasmid was constructed by Abdallah Al-Hakim. pcDNA5 GFP-PLK4, and GFP-PLK4kin + L1 were gifts from Daiju Kitagawa. This work was funded through a Canadian Institute for Health Research Foundation Grant, an ORF RE-5 grant from the Ontario Ministry for Research, and a grant-in-aid from the Krembil Foundation. LP holds a Tier 1 Canada Research Chair in Centrosome Biogenesis and Function. The Network Biology Collaborative Centre at the LTRI is supported by the Canada Foundation for Innovation, the Ontario Government, and Genome Canada and Ontario Genomics (OGI-139).

## Additional information

### Funding

| Funder | Grant reference number | Author |
| --- | --- | --- |
| Krembil Foundation | | Laurence Pelletier |
| Canadian Institutes of Health Research | CIHR FDN 167279 | Laurence Pelletier |
| Ontario Research Foundation | | Laurence Pelletier |

The funders had no role in study design, data collection and interpretation, or the decision to submit the work for publication.

### Author contributions

Johnny M Tkach, Conceptualization, Formal analysis, Investigation, Methodology, Software, Validation, Visualization, Writing - original draft, Writing – review and editing; Reuben Philip, Investigation, Methodology, Validation, Writing – review and editing; Amit Sharma, Investigation, Validation, Writing – review and editing; Jonathan Strecker, Investigation, Methodology, Writing – review and editing; Daniel Durocher, Supervision, Writing – review and editing; Laurence Pelletier, Conceptualization, Formal analysis, Funding acquisition, Methodology, Resources, Supervision, Writing - original draft, Writing – review and editing

### Author ORCIDs

Johnny M Tkach http://orcid.org/0000-0001-6118-9677
Reuben Philip http://orcid.org/0000-0001-7146-5284
Daniel Durocher http://orcid.org/0000-0003-3863-8635
Laurence Pelletier http://orcid.org/0000-0003-1171-4618

### Decision letter and Author response

Decision letter https://doi.org/10.7554/eLife.73944.sa1
Author response https://doi.org/10.7554/eLife.73944.sa2

## Additional files

### Supplementary files
• Supplementary file 1. Summary of screening data.
• Supplementary file 2. Gene enrichment details for RPE-1 screens.

• Supplementary file 3. Reagents used in this study.

• MDAR checklist

• Source data 1. All original Western blot files (.zip file).

• Source data 2. All unaltered original images for Western blot labeled with 1395 relevant bands. (.zip file).

## Data availability

All data generated or analyzed during this study are included in the manuscript and supporting files. Source data and unaltered Western blots have been provided for all Figures. CellProfiler analysis pipelines available from Zenodo (https://zenodo.org/record/6532747).

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
