## [Editor Report]

This study analyses the molecular pathways that lead to growth arrest in human cells with altered centriole number, caused by inhibition of PLK4, the master regulator of centriole duplication. The authors identify the ubiquitin E3 ligase TRIM37 as a key mediator of this growth arrest, but, in contrast to previous work, they find that growth arrest correlates better with PLK4 activity than with the duration of mitosis or centrosome number. The work extends the role for TRIM37 in regulating centrosome protein levels, distribution, and function, and will be of interest to researchers interested in centrosome and growth regulation.

---

## [Decision Letter]

[Editors' note: this paper was reviewed by Review Commons.]

---

## [Author Response]

Reviewer 1, Major points1. Previous data suggested that an important role of TRIM37 was to limit accumulation of CEP192 levels, yet here CEP192 levels appeared unchanged in TRIM37 knockout cells that stably express wild-type or RING domain mutant TRIM37. However, in agreement with previous work, transient expression of TRIM37 reduced CEP192 levels along with those of other PCM and centriole components in an E3-dependent manner. These data are rather confusing in light of the literature, and the current report does not really deal with these discrepancies but to me they suggest that high levels of TRIM37 can target multiple centrosome components for degradation, but this may be an experimental artefact.

We thank the reviewer for pointing this out. To investigate this further, we performed additional replicates for the experiments examining the total cellular levels of PCM components in WT RPE-1, *TRIM37*^-/-^, and *TRIM37*^-/-^ stably overexpressing FLAG-BirA(FB) TRIM37. We now find that stable overexpression of FB-TRIM37 also results in lower steadystate levels of CEP192 (but not PCNT or CEP215). These results are found in Figure 6A and Figure 6B and Figure 6 —figure supplement 1A and *B* and are now consistent with the literature showing that induced overexpression of TRIM37 affects CEP192 levels. However, we still do not observe reciprocal changes (i.e. an increase in abundance of CEP192) after *TRIM37* disruption. We also performed Western blot and immunofluorescence experiments using WT RPE-1 and *TRIM37*^-/-^ cells induced to express TRIM37-3xFLAG or stably expressing FB-TRIM37 to compare overall protein levels and centrosomal levels more directly (Figure 4 —figure supplement 2A-C). We find that induced TRIM37-3xFLAG is expressed at 4- to 5-fold higher levels than the endogenous protein while stably expressed FB-TRIM37 is expressed at 150- to 200-fold higher levels. We discuss that overexpressed TRIM37 may account for some of the observed phenotypes and/or our *TRIM37*^-/-^ clonal line may have acquired a mutation that suppresses the loss of TRIM37 (Page 14, paragraph 1) of the revised manuscript (also see Reviewer 1, major point 4).

In addition, we determined that the amount of CEP192 at mitotic centrosomes decreases after 4 or 8 hours of acute induction of TRIM37 and found that the abundance of another PCM component, PCNT, and a non-PCM centriole component, CEP120 are also diminished at mitotic centrosome after 4 or 8h acute TRIM37 expression. These data indicate that the effect of induced overexpression of TRIM37 is not restricted to CEP192 (Figure 6D). We now further discuss that the exact substrates, or proteins affected by TRIM37 expression or depletion remain to be fully delineated. Further work will be required to delineate direct versus indirect effects of TRIM37 and the mechanisms that control these functions (Page 14, paragraph 1).

2. The choice of cells for particular experiments is not always stated or explained. For instance, in Figure 3A: Trim37 KO pool used while in Figure 3B TRIM37 single KO. These are then combined with both transient and stable expression of TRIM37 mutants.

We apologize for not always stating / explaining this well enough. In principle, however, we find this to be a strength of our work. Our key results can be reproduced using *TRIM37*^-/-^ lines generated by different methods and using different sgRNAs ensuring that we are not observing artefacts due to clonal selection or off-target effects. We first confirmed our screen results by infecting cells with virus to deliver two independent guides against TRIM37. Having confirmed the screen results, we generated a clonal *TRIM37*^-/-^ line. To further ensure the reproducibility of our observations we created additional *TRIM37*^-/-^ pools in two different cell lines. All experiments in our manuscript after the initial confirmation (Figure 3A) are now performed using our clonal TRIM37^-/-^ line except for those in Figure 4B and C and Figure 4 —figure supplement 1D and E, where we use pools for the specific reason of disrupting TRIM37 in an independent manner. This is now more clearly noted in the revised version of the manuscript (Page 8, paragraph 3).

3. Two different concentrations (200 nM and 500 nM) of centrinone were used to compare responses of too many or no centrosomes in RPE1 and A375. While these concentrations result in centrosome amplification (200 nM) and loss (500 nM) in RPE1 cells, the phenotypes seem much less clear-cut in A375 cells. At 200nM 70% of cells have 0 or 1 centrioles (~35% each category) and only about 15% have centrosome amplification, whereas centrosome amplification occurs in 30% of RPE1 with 0-1 centrioles seen in fewer than 10% (Figure 4—figure supplement 1H). Hence the different outcomes of centrinone treatment makes conclusions about cell-type specific responses difficult. This difference may be due to differences in drug uptake/efflux, PLK4 activity or in expression of other components of these pathways. In fact, 167nM centrinone B in A375 cells would have been a much closer match to the 200nM treatment of RPE-1. These points should be discussed as they impact the conclusions.

The reviewer rightly points out that the response to centrinone appears to differ between cell types, as shown previously (Meitinger *et al.*, 2020 and Yeow *et al.*, 2020), and that this difference may impact our conclusions. We concur that 167 nM centrinone B in A375 cells would have been a better match to 200 nM centrinone B in RPE-1 cells based on our centrosome count experiment (Figure 4C and Figure 4 —figure supplement 1E). However, the screen results still reveal similar differential responses to the two centrinone B doses used. This caveat is now explicitly discussed in the revised manuscript (Page 13, paragraph 2). The 200 nM centrinone B screens in both cell lines identified genes known to respond to supernumerary centrosomes (ex. ANKRD26, CASP2, PIDD1, CRADD) and centriole components that when disrupted might decrease centrosome load (ex. STIL, CENPJ). Interestingly, the A375 screen identifies more genes in the latter group than the RPE-1 screen (ex. CEP120, CEP135, CEP152, SASS6; see Figure 2 —figure supplement 2C and F).

4. I find the different outcomes of stable versus acute expression of TRIM37 ligase mutant confusing. Here, stable expression of TRIM37 ligase mutant increases mitotic length compared to that of TRIM37 wild-type, which contradicts a recent report by (Meitinger et al. 2021). What could be the potential reason for these differences?

Our initial experiments were performed using a cell line stably expressing FB-TRIM37 C18R. We performed new experiments to compare mitotic length in *TRIM37*^-/-^ cells induced to express TRIM37 C18R-3xFLAG. Using this system, we find that, like Meitinger *et al.*, 2021, induced TRIM37 C18R-3xFLAG fails to increase mitotic length in the presence of centrinone (Figure 5B). This contrasts with stably overexpressed FB TRIM37 C18R that increases mitotic length that was presented in former Figure 5C. Although the data using the stable cell lines we included in the original manuscript with the rescue mutant is still valid (former Figure 5D), we decided to remove it from the revised manuscript since the inducible system provided more robust results and was used in subsequent experiments. We offer two possible explanations for this observation. First, similar to what we observed with wild type TRIM37, the expression level of FB-TRIM37 C18R is higher than induced TRIM37 C18R-3xFLAG. We discuss that high amounts of the E3-ligase mutant might drive the interaction of a critical complex stabilized by TRIM37 or that the catalytic inactive mutant might interact with a substrate that is normally degraded by wild type TRIM37, thus phenocopying the mitotic arrest phenotype (Page 14, paragraph 2). However, we still find that the induced expression of TRIM37 C18R-3xFLAG partially rescues the growth arrest phenotype in clonogenic assays. We present these new data in Figure 4E and F. To date, the model for growth arrest after PLK4 inhibition has focused on mitotic length. We provide multiple pieces of evidence, including those just described, that challenge this model. We discuss our findings that are inconsistent with a mitotic length model on Page 14, paragraph 3.

What could be the mechanism for TRIM37 action in regulating spindle assembly/mitotic duration and cell proliferation upon centrosome loss? How do those acentrosomal MTOCs form that decrease mitotic duration and promote proliferation?

These are exciting questions that explore details of TRIM37 mechanism that are not addressed in the current manuscript. Our data challenges the view that mitotic length contributes to cell growth arrest mediated by TRIM37. Rather, we highlight the interplay between PLK4, TRIM37 and growth arrest. This does not mean that TRIM37 doesn’t contribute to ectopic mitotic structures or mitotic length, only that mitotic length is perhaps not the sole determinant of growth arrest. This is now better discussed on Page 14, paragraph 4 of the manuscript and included in our working model (Figure 8).

Do the authors find a difference in the % of cells expressing TRIM37 mutants upon stable or acute expression? This part needs a better summary, and again a table would help. I also wonder about protein expression levels; wild-type FB-TRIM37 seems to be expressed at much lower levels than the mutants in Figure 5B.

The differences in overall abundance are not due to heterogenous expression within the population. The TRIM37 mutants are expressed in all cells after stable expression (Figure 3 —figure supplement 2B and C, and Figure 7 —figure supplement 1B and D). Although we did not formally test this in the induced cell lines, these lines are selected using G418. TRIM37 is proposed to negatively regulate its own expression in an E3-dependent manner (Meitinger et al. 2021, Figure 3f). Our results are consistent with this as the TRIM37 C18R and TRIM37 DRING mutants have a higher overall abundance compared to TRIM37 or TRIM37 D505709 using either stable or induced cell lines. Some of the functional differences we initially observed can be explained by stable overexpression of FB-TRIM37 C18R versus induced expression of TRIM37 C18R3xFLAG. We have redone experiments using an inducible system and our results are now more consistent with those in the literature. We provide a table in Figure 7 —figure supplement 2A to summarize the TRIM37 mutants we used and their phenotypes.

5. Other means of centrosome depletion (Cenpj, SAS6 etc) would have been useful to include in the manuscript in support of E3 ligase dependent and independent roles of TRIM37. It is not essential to perform these experiment but if data are available, including these would improve the paper.

To eliminate centrioles using another approach, we created an RPE-1 TRIM37^-/-^ SASS6^-/-^ cell line. Disruption of SASS6 suppresses growth arrest at all centrinone concentrations tested, including 125 nM centrinone B where arrest is largely TRIM37-independent (Figure 4 —figure supplement 3A and B). We find that similar to centriole loss induced by 500 nM centrinone B, TRIM37 C18R-3xFLAG partially rescues growth arrest in these conditions (Figure 4F). These data together suggest that TRIM37 catalytic activity is not strictly required for growth arrest function.

6. The authors show that TRIM37 regulates PLK4 phosphorylation and that this modification could only be observed in HEK293T and not in RPE1. Why would there be a difference between HEK293 and RPE1?

In our original manuscript we did not directly compare 293T and RPE-1 using the same experimental conditions. We were referring to the appearance of PLK4 in our immunoprecipitation experiments using RPE-1 cells (Figure 7B; former Figure 4B) compared to PLK4 observed in (Figure 7D-G; former Figure 7A-D). There are key differences in how these experiments were performed including a longer transfection time and the presence of MG132 using 293T cells that might affect PLK4 modification. We were incorrect to try and make the comparison and apologize for this indiscretion.

However, we have now performed experiments using 293T, HeLa and RPE-1 cells. We find marked PLK4 modification after co-expressing TRIM37 and PLK4 in 293T and HeLa cells. However, such modifications are much less evident in RPE-1 cells.

We speculate that the presence of the modified forms will depend on rates of modification (phosphorylation, ubiquitination or other) and de-modification (dephosphorylation, deubiquitination, etc.) and on any proteins that might also form a complex with PLK4 and TRIM37. Interestingly, 293T and HeLa cells both contain extra copies of chromosome 17 that harbors the *TRIM37* locus. Since we observe modification in at least two of the cell lines tested, we conclude that this is not cell type specific. We present these data in Author response image 1 to the editor and reviewers but did not include them in our revised manuscript.

**Author response image 1. sa2fig1:** TRIM37-dependent PLK4 modification is observed in multiple cell lines. 293T, HeLa and RPE-1 cells were transfected with plasmids to express Myc-PLK4 and one of eGFP, T7-TRIM37 or T7-TRIM37D505-709. Cells were incubated for 48 with 30 µm MG132 added for the final 7 h. Cells were harvested directly in SDS-PAGE sample buffer and prepared for Western blot to detect PLK4. Ponc.S indicates total protein. A representative experiment from 3 Western blots is shown.

7. Statistical analysis for graphs should be included. Figure 5 is ok but graphs in Figures 3, 4, 6, 7 would benefit.

This point is well taken. Statistical analyses have now been performed for the data in these figures as indicated and significant (*p* < 0.05) values clearly noted to support our conclusions.

8. The authors characterise TRIM37 localisation. They detect it at centrosomes (as shown by Yeow et al. 2021) and more specifically at the PCM, but apparently the signal is not present in all cells. They should also provide a quantification of the % of cells with centrosomal TRIM37 signal and compare this to cells expressing Flag-tagged Trim37. The specificity of the antibody signal using TRIM37-/- should be confirmed.

We improved our immunofluorescence protocol to include a pre-extraction step and TRIM37 signal at the centrosome is now evident in a majority of cells. We confirmed the specificity of the antibody using this new protocol by staining TRIM37^-/-^ cells and cells depleted for TRIM37 using siRNA (Figure 3 —figure supplement 1E-G). We found that the TRIM37 signal was significantly decreased at the centrosome in interphase cells and mitotic cells, but the overall decrease was lower in mitotic cells. We therefore restricted our analysis to interphase cells. We also provide a more detailed analysis of TRIM37 localization based on cell cycle and mother/daughter centriole (Figure 3D-G). Finally, we quantify the centrosomal intensity of wild type FB-TRIM37, the E3-ligase mutants C18R and DRING (Figure 3I-J), and the PLK4 interaction mutants 505-709 and D505-709 (Figure 7 —figure supplement 1C and E).

Reviewer 1, Minor points1. Page 3: "A recent screen for mediators of supernumerary centrosome-induced arrest identified PIDDosome/p53 and placed the distal appendage protein ANKRD26 within this pathway [31]". It appears that the reference for Burigotto et al. is missing.

This reference has been inserted on Page 3, paragraph 2 of the revised manuscript.

2. Page 6: The authors state that: TP53BP1, USP28 and CDKN1A are also suppressors in the Nutlin-3a screen and suggest that they act in a general p53 pathway. However Meitinger et al. (2016) showed that depletion of TP53BP1 or USP28 did not affect the upregulation of p53 and p21 upon Mdm2 inhibition.

We first note that we used Nutlin-3a to stabilize p53 while Meitinger *et al.* use a different inhibitor, although they both act by blocking the interaction between p53 and Mdm2. The data from Meitinger *et al.* explore the abundance of p53 and p21 and cell growth after Mdm2 inhibition. 53BP1 and USP28 are thought to affect p53 transcription (Cuella-Martin R *et al.*, 2016). As such, p53 abundance should not be affected. Without quantification it is difficult to assess the relative abundance of the p53 transcriptional target, p21. The abundance of p21 appears to be lower in *USP28*^-/-^ cells, and possibly in *TP53BP1*^-/-^ cells. We note that our data is consistent with a previous report indicating that TP53BP1 and USP28 are required for cell arrest after Nutlin-3a treatment (Cuella-Martin R *et al.* 2016). We, and Cuella-Martin *et al.* use clonogenic assays to assess cell growth in the presence of Nutlin-3a. It is possible that this is a more sensitive assay than serial regrowth performed by Meitinger *et al.* Clonogenic assays allow for continuous growth while slow growth might not be well captured by continuously re-seeding cells. We mention that our results are consistent with those of Cuella-Martin *et al.* (Page 6, paragraph 1), but do not discuss this point further due to space constraints.

3. Page 9: "First, we performed live cell imaging to measure mitotic length in cells grown in centrinone". For consistency the authors should say centrinone B here as well.

We now use ‘centrinone B’ throughout the manuscript.

4. Page 9: "Cells lacking TRIM37 suppressed the growth arrest from 150 to 500 nM centrinone B in RPE-1 and 167 to 500 nM in A375 cells". The growth data for the A375 cells seem to be missing from the figures.

The text for the results has changed significantly to improve the clarity of our manuscript. The growth data for A375 cells is now presented in Figure 4 —figure supplement 1C.

5. Page 10: "Our results confirmed that PLK4 and TRIM37 form a complex in RPE-1 cells (Figure 3G)" It appears the authors referred to the wrong figure, it should be Figure 4B.

Our apologies. The text of the results has changed significantly to improve the clarity of our manuscript. This statement now refers to the correct figure panel (Figure 7B)

6. Figure 1C: The nuclear p53 signal is not apparent with 500 nM centrinone B in the exemplary cells. Did the authors use thresholding to quantify p53/p21 positive cells?

To quantify the data, we used automated image analysis and set a cut off based on p53 intensity in DMSO-treated cells to indicate p53-positive cells. The p53 staining in centrinone-treated cells is variable and we initially chose the same cells to show p53 staining and centrosome number. We improved the figure by showing these data in separate panels. We repeated the experiment and captured a lower magnification image to show a more representative field of cells stained for p53 (Figure 1C) and separate high magnification to show examples of centrosome number (Figure 1D). The quantification pipeline is now better explained in the methods section.

7. Figure 4D and Figure 4—figure supplement 1G: The graph is misleading and should not be presented as a continuous line.

We apologize for this. We have now changed the way this data is presented to make it easier to understand and to facilitate indicating statistical differences (Figure 4B and Figure 4 —figure supplement 1C). Instead of a scatter plot of all the data, we now present the data from individual replicates as open circles with the mean and standard deviation at each centrinone B concentration with statistical differences indicated. We hope this addresses any confusion regarding these data.

8. Figure 5A and C: A direct and statistical comparison mitotic timing upon expression different Trim37 mutants to wildtype and trim37-/- cells is missing

In Figure 5A we compare RPE-1 WT to TRIM37^-/-^ at each centrinone B concentration and within each line we compare each centrinone B concentration to DMSO. Perhaps we do not understand the reviewer’s concern here, but we do not think any comparisons are missing from the data panel. The data originally in Figure 5C has been removed and replaced by the data in Figure 5B. Appropriate statistical tests were made among the samples. We hope this addresses this concern.

9. Figure 6B: A loading control/Ponceau staining is missing as well as the quantification of protein levels

This experiment was repeated for proper quantification with an appropriate loading control for our representative results (Figure 6B and Figure 6 —figure supplement 1B).

10. Figure 6D: It is unclear if the centrosomal signal intensity was quantified in interphase or mitotic cells

The centrosomal signal was quantified in mitotic cells only. This results and figure legend have been updated to more clearly indicate this.

11. Figure 7C: A loading control/Ponceau staining is missing

The experiment was repeated and a Coomassie Blue stained membrane is provided to indicate the input amounts for each sample (Figure 7F).

12. Figure 2—figure supplement 2F and G: It would help if the authors could highlight the cell line, e.g. RPE-1 (F) or A375 (G) in the venn diagrams.

In Figure 2 —figure supplement 2F we now highlight the genes found in RPE-1 and A375 screens only in the overlap of the Venn diagram using font colour. We now colour code the hits from each cell line in panels (F) and (G). We thank the reviewer for this suggestion.

13. Figure 4—figure supplement 1E: it appears that the BirA antibody gives only an unspecific signal. It would be useful to show if the different TRIM37 variants are able to localise to the centrosomes. Furthermore it appears that centrosomes are missing in the C18R and 505-709 variants. It would be useful if the authors quantify centrosome numbers upon expression of different Trim37 variants as shown in Figure 4—figure supplement 1. To make the identification of the cell easier it would help to include a DNA signal or indicate the outline of the cell.

The anti-BirA antibody does give a diffuse signal, although we feel it is specific considering that the BirA signal is only observed in cells expressing FLAG-BirA alone or BirA fusion proteins.

Having said this, we have changed the way we present this data. For the stable rescue lines generated using the RPE-1 TRIM37^-/-^ clone, we present detailed characterization including percentage of cells expressing the fusion protein using the BirA antibody (Figure 3 —figure supplement 2B and C; Figure 7 —figure supplement 1B and D) and the total centrosomal intensity using FLAG or TRIM37 antibodies (Figure 3I and J; Figure 7 —figure supplement 1C and E). We provide Western blots for the stable expression of the fusion proteins in RPE1 and A375 TRIM37^-/-^ pools in Figure 4 —figure supplement 1A and B.

For images showing centrosomes and ectopic centrosomal structures/Cenpas, we used a closed arrowhead to indicate the centrosome and caret mark (open arrowhead) to indicate the Cenpas structure (Figure 6F; Figure 6 —figure supplement 2A-C). Additionally, these phenotypes are quantified to show penetrance (Figure 6 —figure supplement 2E). We hope this improves our presentation in these figure panels.

14.The generation of stable and dox-inducible cell lines is missing in the material and methods

We apologize for this omission. This information has now been added.

Reviewer 2, Major points1. The centrosomal localization of endogenous TRIM37 should be validated by comparing control and knockout/knockdown cells.

We have performed these experiments as outlined in response to Reviewer 1, Major point 8.

We improved our immunofluorescence protocol to include a pre-extraction step and TRIM37 signal at the centrosome is now evident in a majority of cells. We confirmed the specificity of the antibody using this new protocol by staining *TRIM37*^-/-^ cells and cells depleted for TRIM37 using siRNA (Figure 3 —figure supplement 1E-G). We found that the TRIM37 signal was significantly decreased at the centrosome in interphase cells and mitotic cells, but the overall decrease was lower in mitotic cells. The general cytoplasmic signal was also qualitatively decreased although we did not measure the signal in this compartment. We therefore restricted our analysis to interphase cells.

2. Some of the quantifications are derived from only two experiments and in many cases no statistical testing was done. The authors should test the observed effects and add extra replicates to make the data more robust, where required.

This issue was also raised by Reviewer #1, major point 7. We added replicate experiments where required and statistical analyses performed are indicated with significant (*p* < 0.05) values included to support our conclusions..

3. Figure 5 supplements: panels showing effects on marker proteins in cells by IF lack quantification of the claimed effects. Without providing some type of quantifications for key findings, it is unclear how strong or penetrant the effects are.

Figure 5 —figure supplement 1C has been removed. We provide quantification for former Figure 5 —figure supplement 1D, now Figure 6 —figure supplement 2C and D. Data replacing former Figure 5 —figure supplement 2A and B has been quantified and is now displayed in Figure 5C and D. See also comment to Reviewer 1, Major point 7.

Reviewer 2, Minor pointsI would suggest a final, summarizing schematic that illustrates the main findings in a cartoon/flow chart manner.

We have extensively reworked the discussion of our main findings, provided a summary table in Figure 7 —figure supplement 2A, and provided a summary model in Figure 8 to increase the clarity of our manuscript.

1. Please revise incorrect abstract sentence: "We identify TRIM37 as a key mediator of growth arrest when PLK4 activity is partially or fully inhibited but is not required for growth arrest triggered by supernumerary centrosomes."

In our screens, we find that TRIM37 is required for growth arrest after treating cells with 200 and 500 nM centrinone B. Treatment of cells with 200 nM centrinone B causes centriole overduplication and our initial hypothesis was that centriole overduplication alone is inducing growth arrest. To test this in a parallel manner, we also overexpressed PLK4 to induce centriole overduplication. Surprisingly, but consistent with recently published results (Evans *et al.*, 2020), TRIM37 was not required for growth arrest after PLK4 overexpression (*Figure 4G*). Thus, TRIM37 is required for growth arrest after 200 nM centrinone treatment, but not PLK4 overexpression, yet both conditions result in centriole overduplication. This concept is further highlighted, discussed and clarified in the discussion (*Page 13, paragraph 3*). We have also removed the confusing sentence in the abstract since it was rewritten anew during the revision process.

Please also see similar comment to Reviewer 3, Major point 1.

2. In various figures and supplements showing centrosome and condensates/Cenpas, these are very difficult to distinguish due to their small size. I suggest to magnify regions of interest and/or add arrowheads in different colors marking the specific structures.

This comment is similar to Reviewer 1, Minor point 13. We remade the image panels, cropping as tightly to the desired structures as possible and enlarged the images as much as we could to fit the figure panel. Additionally, we used arrowheads to indicate centrosomes and caret marks (open arrowheads) to indicate condensates/Cenpas (Figure 6F; Figure 6 —figure supplement 2A-C). Additionally, these phenotypes are quantified to show penetrance (Figure 6 —figure supplement 2E). We hope this improves the clarity of these figure panels.

3. Figure 2A: What is the purpose of the schematics on the right of panel A? The labels in the graph are unreadable and the network diagram without any labels is also not very useful. This could be removed.

The schematics on the right indicate a ‘generic analysis’ using the NGS sequencing data. We agree it is not essential and have removed it the revised version of the manuscript.

4. Figure 2B: The network presentation is not very easy to read. What are the functional groups/pathways here? The clusters should be labeled accordingly. What is the meaning of the different sizes of the circles? Maybe key interactions (e.g. TRIM37) could be indicated in a different color shade to highlight these?

In our figure we tried to highlight (1) the connectivity among screening conditions and (2) complexes that were identified by the screens. Each node (other than the six hub nodes that denote a screen condition) represents a hit from the screens. Thus, the nodes are connected by edges only to the screening conditions, not to each other. In this scenario, highlighting TRIM37 ‘interactions’ would only highlight the screening conditions for which TRIM37 was a hit (200 nM RPE-1, 500 nM RPE-1, 200 nM A375, 500 nM A375). We could try to overlay functional enrichment data on the graph, but this data is presented separately in Figure 2 —figure supplement A-D. To distinguish the ‘hub’ nodes, representing the screen conditions, from the others we have made them larger and are outlined with yellow. The large circles represent hits found in previous PLK4 inhibition screens; we now indicate this both in the legend in the figure panel (Figure 2B) and in the written figure legend (Page 26).

Reviewer 3, Major points1. The presentation throughout the manuscript sometimes made it difficult to follow exactly what the authors meant when they referred to the various doses of Centrinone used in their experimentsoften using the terms "low" or "high" without specifying exactly what they mean. In Figure 1A, for example, they present a growth inhibition curve using a log10 scale of Centrinone concentration, and they conclude that growth was inhibited "at concentrations above 150nM, with full inhibition observed at concentrations greater than 200nM". I presume this is just sloppy language, as it appears that growth is significantly inhibited at 150nM and full growth inhibition is achieved at 200nM. However, in Figure 4D, the authors show another growth inhibition curve (this time presented on a linear scale) where significant growth inhibition is seen well below 100nM and full inhibition appears to be achieved at ~125nM. The discrepancy between these experiments is not noted, nor any reason for it explained.

We agree with the reviewers and apologize for using ‘low’ and ‘high’ as they are ambiguous. We now refer to specific centrinone B concentrations or to inhibition ‘phases’ based on TRIM37-dependent growth arrest and centrosome number. The direct comparison between former Figure 1A and former Figure 4D (now Figure 4B) is not straightforward. The experiments presented were performed approximately 6 years apart and in slightly different ways. As reviewer 3 indicates, Figure 1A is presented in a log scale; this makes it difficult for the reader to determine the exact concentrations of centrinone B used. For this panel, we used, 0 (DMSO), 10, 30, 75, 165, 200 and 500 nM centrinone B. For former Figure 4D, we used 0, 50, 125, 150, 167, 200 and 500 nM. The only point that might be anomalous is 75 nM in Figure 1A. We do see approximately 25% inhibition using 50 nM centrinone B in Figure 4D, but no inhibition using 75 nM in Figure 1A. We can offer two explanations for this discrepancy. First, we noticed small deviations in the potency of centrinone B batches. Second, for Figure 1A, cells were assayed using a passaging assay where they are continuously plated, counted and re-seeded. Cells in Figure 4D were assayed using a clonogenic assay where cells are plated at low density and allowed to grow over the course of approximately two weeks. It is possible that a combination of these factors led to the discrepancy pointed out by the reviewer. To streamline this aspect of the manuscript, we have made the following change: we repeated the growth experiment for Figure 1A and now present the data by plotting the results from the individual replicates with the mean and standard deviation. We show only the 200 and 500 nM centrinone B concentrations since we derive our basic conclusions from these conditions (i.e. both concentrations result in p53dependent growth arrest where centrioles are overduplicated after 200 nM centrinone B, while centrioles are lost after treatment with 500 nM) and these are the conditions we subsequently used in our screens. We hope that this explanation and changes satisfy this reviewer.

While discrepancies such as this may seem trivial, they make it hard to interpret some of the authors conclusions. For example, in their initial screen, the "low" dose of Centrinone (200nM) leads to centriole amplification and genes that block centriole duplication or PIDDosome function (which normally signals the presence of extra centrioles) are required for the growth arrest triggered by this concentration of the drug (Figure 1B). To me, this suggests that centriole amplification is required for this growth arrest at 200nM. However, when the authors test a more graded series of concentrations they conclude "excess centrioles might not be the trigger for this arrest at low Centrinone B concentrations". I assume they are using "low" here to indicate concentrations at or below 150nM (even though they use low to mean 200nM in their initial screen)? In the Discussion, they state that TRIM37 is "required for the growth arrest in response to partially or fully inhibited PLK4, but this activity was independent of the presence of excess centrioles". Again, it is not clear to which experiments they are referring when they talk about "partially" or "fully" inhibited PLK4, but, if this is correct, then why are genes required for centriole duplication and PIDDosome function identified in their initial screen as being required for the growth arrest at 200nM but not 500nM? Do they consider 200nM to be fully inhibiting PLK4?

We apologize for not being clearer about these points. We now characterize the doseresponse to centrinone B based on growth arrest, centrosome number and TRIM37 dependence (Page 8, paragraph 3, Figure 4E*).* Briefly, we observe cell arrest in the following phases: (1) without major centrosome abnormalities at 125 nM centrinone B (Phase I); (2) with supernumerary centrosomes at 150-200 nM centrinone B (Phase IIa); (3) with centrosome loss at 500 nM (Phase IIb). Phase I growth arrest is independent of TRIM37 while Phase II is dependent on TRIM37. Moreover, where required, we refer to specific concentrations of centrinone B, rather than use the relative terms ‘low’ and ‘high’.

We initially observed that cells arrested after treatment with either 200 or 500 nM centrinone B. Additionally, we observed centriole over-duplication in cells treated wtih 200 nM but centriole loss with 500 nM. Our initial hypothesis was that either centriole overduplication or loss resulted in growth arrest. To determine if centriole overduplication caused by 200 nM centrinone B triggers growth arrest, we induced centriole overduplication using a parallel method by overexpressing PLK4 and, surprisingly, TRIM37 was not required for growth arrest in these conditions, similar to that observed by Evans *et al.*, 2020. Thus, there are two conditions where centriole overduplication is observed and the growth arrest in only one condition is dependent on TRIM37. This is an important difference that we now better highlight in our revised manuscript on Page 13, paragraph 3. We now present a model (new Figure 8) and a table (Figure 7 —figure supplement 2A) that summarizes our results.

Briefly, it is thought that partially inhibited PLK4 blocks its own auto-phosphorylation and therefore blocks its degradation. The overall abundance of PLK4 therefore increases under these conditions and overduplication occurs. In our hands, we consider PLK4 to be partially inhibited in RPE-1 or A375 cells at any concentrations of centrinone B at 200 nM or lower.

We agree that it is interesting we detect genes required for centriole duplication and PIDDosome function in our 200 nM screen. We now discuss this more explicitly. We note that TRIM37 suppression of growth arrest is only partial, suggesting another pathway might be active. We suggest that at 200 nM there is a centrosome-dependent response characterized by the ANKRD26/PIDDosome pathway and a PLK4 activity pathway that TRIM37 participates in. We compare our screen results to those of Evans et al. to identify genes potentially involved in these two different responses (Page 13, paragraph 3).

Please also see similar comment to Reviewer 2, Minor point 1.

Presumably it will only require textual changes to address this point, but it is hard to assess the broader significance of the paper until these points are clarified: is the main point of this paper that the cells response to Centrinone treatment is complicated and the role of TRIM37 equally so; or, is there a narrative that leads to a clear hypothesis that can explain these surprising findings?

The current model suggests that TRIM37 E3-dependent remodeling of CEP192 and

subsequent mitotic length changes underlies its growth arrest activity after centriole loss. We have data that is consistent with previously published results and data that challenges some of these recent reports. Importantly, our data does not support a model where mitotic length is the sole determinant of growth arrest and provide a new model for growth arrest and TRIM37 function (Figure 8). While revising our manuscript we performed an additional set of experiments to test this by indirectly monitoring PLK4 activity after centrinone B treatment in the presence or absence of TRIM37. To do so, we expressed GFPPLK4kin+L1 (containing the PLK4 kinase domain plus the unstructured L1 region or full length GFP-PLK4; see Yamamoto S *et al.*, 2019) and performed FRAP analyses (Figure 4D; Figure 7 —figure supplement 1H and I). In this assay, PLK4 mobility is linked to its phosphorylation status near the phosphodegron where decreased mobility corresponds decreased phosphorylation. We find that PLK4 mobility decreases in a dose-dependent manner after treatment with centrinone B mirroring the cell arrest phenotype leading us to suggest cell arrest is directly dependent on PLK4 activity. We hypothesize that PLK4, or a substrate of PLK4 is important for the growth arrest signal (Figure 8).

2. It seems a striking omission that the authors show that p53 and p21 are induced by 200nM and 500nM Centrinone (Figure 1D), but they don't assay these proteins at any concentration lower than this. Perhaps they are saving this data for a subsequent manuscript, but the authors certainly seem to draw conclusions from several experiments they perform at concentrations below 200nM, so they should at least explain why they don't assay p53 and p21 status in these experiments.

We apologize for not including this data in the original version of the manuscript. We find that p53 and p21 increase in a dose-dependent manner and that this response is attenuated in *TRIM37*^-/-^ cells. We now include that data in Figure 4 —figure supplement 1D.

Reviewer 3, Minor points1. In the abstract the authors claim that the way in which altered centrosome numbers cause a p53-dependent growth arrest is evolutionarily conserved. This is misleading, as it implies that the loss and gain of centrosomes trigger the same arrest (which is probably not correct), and most of the data to date suggests that flies and worms (two popular models for centrosome research) do not have such a growth-arrest pathway.

This is a good point. We modified this statement to indicate that p53-dependent arrest is confined to mammalian cells: “Altered centrosome numbers in mouse and human cells cause p53-dependent growth arrest through poorly defined mechanisms”.

Reviewer 3, comment in ‘significance’I could not discern, however, whether one could draw any broader conclusions than this, in part due to the presentation problems described above. Moreover, in the abstract the authors propose that altering PLK4 activity alone is sufficient to signal growth arrest. This would be an important conclusion, and I presume this refers to the very low dosage Centrinone experiments that trigger growth arrest without altering centrosome numbers and which does not require TRIM37? If so, this arrest is poorly characterised here and will be the subject of a future investigation, so it seems to strange to have this as a major conclusion in the abstract.

We agree. Since we now have some evidence to support this model we have included this information in the abstract and it is now part of the model we now include in Figure 8 (see also our response to Major point 1) and discussed on Page 14, paragraph 4.

References

Meitinger, Franz et al. “TRIM37 controls cancer-specific vulnerability to PLK4 inhibition.” *Nature* vol. 585,7825 (2020): 440-446. doi:10.1038/s41586-020-2710-1

Yeow, Zhong Y et al. “Targeting TRIM37-driven centrosome dysfunction in 17q23-amplified breast cancer.” *Nature* vol. 585,7825 (2020): 447-452. doi:10.1038/s41586-020-2690-1

Cuella-Martin, Raquel et al. “53BP1 Integrates DNA Repair and p53-Dependent Cell Fate Decisions via Distinct Mechanisms.” *Molecular cell* vol. 64,1 (2016): 51-64.

doi:10.1016/j.molcel.2016.08.002

Evans, Lauren T et al. “ANKRD26 recruits PIDD1 to centriolar distal appendages to activate the PIDDosome following centrosome amplification.” *The EMBO journal* vol. 40,4 (2021):

e105106. doi:10.15252/embj.2020105106

Yamamoto, Shohei, and Daiju Kitagawa. “Self-organization of Plk4 regulates symmetry breaking in centriole duplication.” *Nature communications* vol. 10,1 1810. 18 Apr. 2019, doi:10.1038/s41467-019-09847-x